# Optimum-statistical Collaboration Towards General and Efficient Black-box Optimization

**Wenjie Li**                                                                  *li3549@purdue.edu*
*Department of Statistics, Purdue University*

**Chi-Hua Wang**                                                          *chihuawang@ucla.edu*
*Department of Statistics, University of California, Los Angles*

**Guang Cheng**                                                          *guangcheng@ucla.edu*
*Department of Statistics, University of California, Los Angles*

**Qifan Song**                                                               *qfsong@purdue.edu*
*Department of Statistics, Purdue University*

**Reviewed on OpenReview:** *https://openreview.net/forum?id=ClIcmwdIxn*

## Abstract

In this paper, we make the key delineation on the roles of resolution and statistical uncertainty in hierarchical bandits-based black-box optimization algorithms, guiding a more general analysis and a more efficient algorithm design. We introduce the *optimum-statistical collaboration*, an algorithm framework of managing the interaction between optimization error flux and statistical error flux evolving in the optimization process. We provide a general analysis of this framework without specifying the forms of statistical error and uncertainty quantifier. Our framework and its analysis, due to their generality, can be applied to a large family of functions and partitions that satisfy different local smoothness assumptions and have different numbers of local optimums, which is much richer than the class of functions studied in prior works. Our framework also inspires us to propose a better measure of the statistical uncertainty and consequently a variance-adaptive algorithm VHCT. In theory, we prove the algorithm enjoys rate-optimal regret bounds under different local smoothness assumptions; in experiments, we show the algorithm outperforms prior efforts in different settings.

## 1 Introduction

Black-box optimization has gained more and more attention nowadays because of its applications in a large number of research topics such as tuning the hyper-parameters of optimization algorithms, designing the hidden structure of a deep neural network, and resource investments (Li et al., 2018; Komljenovic et al., 2019). Yet, the task of optimizing a black-box system often has a limited budget for evaluations due to its expensiveness, especially when the objective function is nonconvex and can only be evaluated by an estimate with uncertainty (Bubeck et al., 2011b; Grill et al., 2015). Such limitation haunts practitioners' deployment of machine learning systems and invites scientists' investigation for the authentic roles of resolution (optimization error) and uncertainty (statistical error) in black-box optimization. Indeed, it raises a question of optimum-statistical trade-off: **how can we better balance *resolution* and *uncertainty* along the search path to create efficient black-box optimization algorithms?**

Among different categories of black-box optimization methods such as Bayesian algorithms (Shahriari et al., 2016; Kandasamy et al., 2018) and convex black-box algorithms (Duchi et al., 2015; Shamir, 2015), this paper focuses on the class of hierarchical bandits-based optimization algorithms introduced by (Auer et al.,

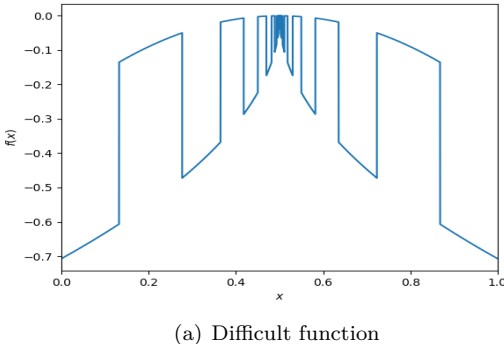
(a) Difficult function

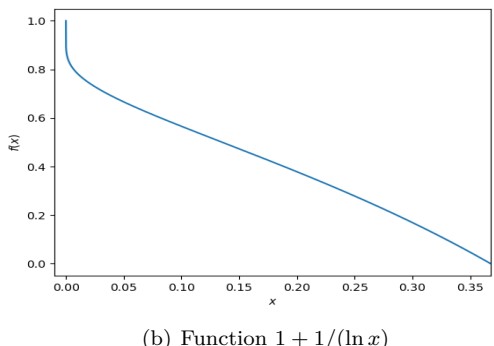
(b) Function $1 + 1/(\ln x)$

Figure 1: 1(a): A difficult function proposed by Grill et al. (2015) which has exponentially increasing $(2\nu_1\rho^h)$-near-optimal regions in the standard partition. 1(b): An example function that violates the $\nu_1\rho^h$ local smoothness assumption by Grill et al. (2015) in the standard partition and thus cannot be analyzed by prior works, but has no local optimum.

2007; Bubeck et al., 2011b). These algorithms search for the optimum by traversing a hierarchical partition of the parameter space and look for the best node inside the partition. Existing results, such as Bubeck et al. (2011b); Grill et al. (2015); Shang et al. (2019); Bartlett et al. (2019); Li et al. (2022), heavily rely on some specific assumptions of the smoothness of the blackbox objective and the hierarchical partition. However, their assumptions are only satisfied by a small class of functions and partitions, which limits the scope of their analysis. To be more specific, existing studies all focus on optimizing "exponentially-local-smooth" functions (see; Eqn. (3)), which can have an exponentially increasing number of sub-optimums as the parameter space partition proceeds deeper (Grill et al., 2015; Shang et al., 2019; Bartlett et al., 2019). For instance, Grill et al. (2015) designed a difficult function (shown in Figure 1(a)) that can be optimized by many existing algorithms because it satisfies the exponential local-smooth assumption. However, functions and partitions that do not satisfy exponential local-smoothness, but with a bounded or polynomially increasing number of near-optimal regions have been overlooked in the current literature of black-box optimization. A simple example is Figure 1(b), which is not exponentially smooth but has a trivial unique optimum. Such a simple example depresses all previous analyses in existing studies due to their dependency on the exponential local-smoothness assumption. What is worse, different designs of the uncertainty quantifier can generate different algorithms and thus may require different analyses. Consequently, a more unified theoretical framework to manage the interaction process between the optimization error flux and the statistical error flux is desirable and beneficial towards general and efficient black-box optimization.

In this work, we deliver a generic perspective on the optimum-statistical collaboration inside the exploration mechanism of black-box optimization. Such a generic perspective holds regardless of the local smoothness condition of the function or the design of uncertainty quantification, generalizing its applicability to a larger class of functions (e.g., Figure 1(b)) and algorithms with different uncertainty quantification methods. Our analysis for the proposed general algorithmic framework only relies on mild assumptions. It allows us to analyze functions with different levels of smoothness and also inspired us to propose a variance-adaptive black-box algorithm VHCT.

In summary, our contributions are:

- We identify two decisive components of exploration in black-box optimization: the resolution descriptor (Definition 1) and the uncertainty quantifier (Definition 2). Based on the two components, we introduce the optimum-statistical collaboration (Algorithm 1), a generic framework for collaborated optimism in hierarchical bandits-based black-box optimization.

- We provide a unified analysis of the proposed framework (Theorem 3.1) that is independent of the specific forms of the resolution descriptor and the uncertainty quantifier. Due to the flexibility of the resolution descriptor, this analysis includes all black-box functions who satisfy the general local smoothness assump-

tion (Condition GLS) and have a finite near-optimality dimension (Definition 1), which are excluded from prior works.

- Furthermore, the framework inspires us to propose a better uncertainty quantifier, namely the variance-adaptive quantifier (`VHCT`). It leads to effective exploration and advantages our bandit policy by utilizing the variance information learnt from past samplers. Theoretically, we show that the proposed framework secures different regret guarantees in the face of different smoothness assumptions, and `VHCT` leads to a better convergence when the reward noise is small. Our experiments validate that the proposed variance adaptive quantifier is more efficient than the existing anytime algorithms on various objectives.

**Related Works.** Pioneer bandit-based black-box optimization algorithms such as `HOO` (Bubeck et al., 2011b) and `HCT` (Azar et al., 2014) require complicated assumptions of both the the black-box objective and the parameter partition, including weak lipschitzness assumption. Recently, Grill et al. (2015) proposed the exponential local smoothness assumption (Eqn. (3)) to simplify the set of assumptions used in prior works and proposed `POO` to meta-tune the smoothness parameters. Some follow-up algorithms such as `GPO` (Shang et al., 2019) and `StroquOOL` (Bartlett et al., 2019) are also proposed. However, both `GPO` and `StroquOOL` require the budget number beforehand, and thus they are not anytime algorithms (Shang et al., 2019; Bartlett et al., 2019). Also, the analyses of these algorithms all rely on the exponential local smoothness assumption (Grill et al., 2015).

## 2 Preliminaries

**Problem Formulation**. We formulate the problem as optimizing an implicit objective function $f : \mathcal{X} \mapsto \mathbb{R}$, where $\mathcal{X}$ is the parameter space. The sampling budget (number of evaluations) is denoted by an *unknown* constant $n$, which is often limited when the cost of evaluating $f(x)$ is expensive. At each round (evaluation) $t$, the algorithm selects a $x_t \in \mathcal{X}$ and receives an stochastic feedback $r_t \in [0, 1]$ , modeled by

$$r_t \equiv f(x_t) + \epsilon_t,$$

where the noise $\epsilon_t$ is only assumed to be mean zero, bounded by $[-\frac{b}{2}, \frac{b}{2}]$ for some constant $b > 0$, and independent from the historical observed algorithm performance and the path of selected $x_t$'s. Note that the distributions of $\epsilon_t$ are not necessarily identical. We assume that there exists at least one point $x^* \in \mathcal{X}$ such that it attains the global maximum $f^*$, i.e., $f^* \equiv f(x^*) \equiv \sup_{x \in \mathcal{X}} f(x)$. The goal of a black-box optimization algorithm is to gradually find $x_n$ such that $f(x_n)$ is close to the global maximum $f^*$ within the limited budget.

**Regret Analysis Framework.** We measure the performance of different algorithms using the *cumulative regret*. With respect to the optimal value $f^*$, the *cumulative regret* is defined as

$$R_n \equiv nf^* - \sum_{t=1}^{n} r_t.$$

It is worth noting that an alternative measure of performance widely used in the literature (e.g., Shang et al., 2019; Bartlett et al., 2019) is the *simple regret* $S_t \equiv f^* - r_t$. Both simple and cumulative regrets measure the performance of the algorithm but are from different aspects. The former one focuses on the convergence of the algorithm's final round output, and the latter cares about the overall loss during the whole algorithm training. The cumulative regret is useful in scenarios such as medical trials where ill patients are included in the each run and the cost of picking non-optimal treatments for all subjects shall be measured. This paper chooses to study the cumulative regret, while in the literature, there were discussions on the relationship between these two (Bubeck et al., 2011a).

**Hierarchical partitioning.** We use the hierarchical partitioning $\mathcal{P} = \{\mathcal{P}_{h,i}\}$ to discretize the parameter space $\mathcal{X}$ into nodes, as introduced by Munos (2011); Bubeck et al. (2011b); Valko et al. (2013). For any non-negative integer $h$, the set $\{\mathcal{P}_{h,i}\}$ partitions the whole space $\mathcal{X}$. At depth $h = 0$, a single node $\mathcal{P}_{0,1}$ covers the entire space. Every time we increase the level of depth, each node at the current depth level will be separated into two children; that is, $\mathcal{P}_{h,i} = \mathcal{P}_{h+1,2i-1} \cup \mathcal{P}_{h+1,2i}$. Such a hierarchical partition naturally

inspires algorithms which explore the space by traversing the partitions and selecting the nodes with higher rewards to form a tree structure, with $\mathcal{P}_{0,1}$ being the root. We remark that the binary split for each node we consider in this paper is the same as in the previous works such as Bubeck et al. (2011b); Azar et al. (2014), and it would be easy to extend our results to the K-nary case (Shang et al., 2019). Similar to Grill et al. (2015), we refer to the partition where each node is split into regular, same-sized children as the standard partitioning.

Given the objective function $f$ and hierarchical partition $\mathcal{P}$, we introduce a generalized definition of near-optimality dimension, which is a natural extension of the notion defined by Grill et al. (2015).

**Near-optimality dimension.** For any positive constants $\alpha$ and $C$, and any function $\xi(h)$ that satisfies $\forall h \geq 1, \xi(h) \in (0,1]$, we define the near-optimality dimension $d = d(\alpha, C, \xi(h))$ of $f$ with respect to the partition $\mathcal{P}$ and function $\xi(h)$ as

$$d \equiv \inf\{d' > 0 : \forall h \geq 0, \mathcal{N}_h(\alpha\xi(h)) \leq C\xi(h)^{-d'}\} \tag{1}$$

if exists, where $\mathcal{N}_h(\epsilon)$ is the number of nodes $\mathcal{P}_{h,i}$ on level $h$ such that $\sup_{x \in \mathcal{P}_{h,i}} f(x) \geq f^* - \epsilon$.

In other words, for each $h > 0$, $\mathcal{N}_h(\alpha\xi(h))$ is the number of near-optimal regions on level $h$ that are $(\alpha\xi(h))$-close to the global maximum so that any algorithm should explore these regions. $d = d(\alpha, C, \xi(h))$ controls the polynomial growth of this quantity with respect to the function $\xi(h)^{-1}$. It can be observed that this general definition of $d$ covers the near optimality dimension defined in Grill et al. (2015) by simply setting $\xi(h) = \rho^h$ and the coefficient $\alpha = 2\nu$ for some constants $\nu > 0$ and $\rho \in (0,1)$.

The rationale of introducing the generalized notion $\xi(h)$ is that, although the number of nodes in the partition grows exponentially when $h$ increases, the number of near-optimal regions $\mathcal{N}_h(\epsilon)$ of the objective function $f$ may not increase as fast, even if the near-optimal gap $\epsilon$ converges to 0 slowly. The particular choice of $\xi(h) = \rho^h$ in Grill et al. (2015) indicates that $\mathcal{N}_h(\alpha\rho^h) \leq C\rho^{-dh}$, which may be over-pessimistic and makes it a non-ideal setting for analyzing functions that change rapidly and don't have exponentially many near-optimal regions.

Such a generalized definition becomes extremely useful when dealing with functions that have different local smoothness properties, and therefore our framework can successfully analyze a much larger class of functions. We establish our general regret bound based on this notion of near-optimality dimension in Theorem 3.1.

It is worth mentioning that taking a slowly decreasing $\xi(h)$, although reduces the number of near-optimal regions, does not necessarily imply that the function is easier to optimize. As will be shown in Section 3 and 4, $\xi(h)$ is often taken to be the local smoothness function of the objective. A slowly decreasing $\xi(h)$ makes the function much more unsmooth than a function with exponential local smoothness, and hence is still hard to optimize.

**Additional Notations**. At round $t$, we use $H(t)$ to represent the maximum depth level explored in the partition by an algorithm. For each node $\mathcal{P}_{h,i}$, we use $T_{h,i}(t)$ to denote the number of times it has been pulled and $r^k(x_{h,i})$ to denote the $k$-th reward observed for the node, evaluated at a **pre-specified** $x_{h,i}$ within $\mathcal{P}_{h,i}$, which is the same as in Azar et al. (2014); Shang et al. (2019). Note that in the literature, it is also considered that $x_{h,i}$ follows some distribution supported on $\mathcal{P}_{h,i}$, e.g., Bubeck et al. (2011b).

## 3 Optimum-statistical Collaboration

This section defines two decisive quantities (Resolution Descriptor and Uncertainty Quantifier) that play important roles in the proposed optimum-statistical collaboration framework. We then introduce the general optimum-statistical collaboration algorithm and provide its theoretical analysis.

**Definition 1. (Resolution Descriptor OE)**. Define $\mathtt{OE}_h$ to be the *resolution* for each level $h$, which is a function that bounds the change of $f$ around its global optimum and measures the current optimization error, i.e., for any global optimum $x^*$,

$$\forall h \geq 0, \forall x \in \mathcal{P}_{h,i_h^*}, f(x) \geq f(x^*) - \mathtt{OE}_h, \tag{OE}$$

where $\mathcal{P}_{h,i_h^*}$ is the node on level $h$ in the partition that contains the global optimum $x^*$.

---

**Algorithm 1 Optimum-Statistical Collaboration (OSC)**

---

1: **Input:** partition $\mathcal{P}$, resolution descriptor $\mathtt{OE}_h$, uncertainty quantifier $\mathtt{SE}_{h,i}(T,t)$, selection policy $\pi(\mathcal{S})$
2: **Initialize** $\mathcal{T} = \{\mathcal{P}_{0,1}, \mathcal{P}_{1,1}, \mathcal{P}_{1,2}\}$
3: **for** $t = 1$ to $n$ **do**
4:     $\mathcal{S} = \{\mathcal{P}_{0,1}\}, \mathcal{P}_{h_t,i_t} = \mathcal{P}_{0,1}$
5:     **while** $\mathtt{OE}_{h_t} \geq \mathtt{SE}_{h_t,i_t}(T,t)$ **do**
6:         $\mathcal{S} = \mathcal{S} \setminus \{\mathcal{P}_{h_t,i_t}\} \bigcup \{\mathcal{P}_{h_t+1,2i_t-1}, \mathcal{P}_{h_t+1,2i_t}\}$
7:         $\pi(\mathcal{S})$ selects a new node $\mathcal{P}_{h_t,i_t}$ from $\mathcal{S}$
8:     **end while**
9:     Pull $\mathcal{P}_{h_t,i_t}$ and update $\mathtt{SE}_{h_t,i_t}(T,t)$
10:    **if** $\mathtt{OE}_{h_t} \geq \mathtt{SE}_{h_t,i_t}(T,t)$ and $\mathcal{P}_{h_t+1,2i_t} \notin \mathcal{T}$ **then**
11:        $\mathcal{T} = \mathcal{T} \bigcup \{\mathcal{P}_{h_t+1,2i_t-1}, \mathcal{P}_{h_t+1,2i_t}\}$
12:    **end if**
13: **end for**

---

**Definition 2. (Uncertainty Quantifier $\mathtt{SE}$)**. Let $\mathtt{SE}_{h,i}(T,t)$ be the *uncertainty estimate* for the node $\mathcal{P}_{h,i}$ at time $t$, which aims to bound the statistical estimation error of $f(x_{h,i})$, given $T$ pulled values from this node. Recall that $T_{h,i}(t)$ is the number of pulls for node $\mathcal{P}_{h,i}$ until time $t$, and let $\widehat{\mu}_{h,i}(t)$ be the online estimator of $f(x_{h,i})$, we expect that $\mathtt{SE}$ ensures $\sum_{t=1}^{\infty} \mathbb{P}(\mathcal{A}_t^c) < C$ for some constant $C$, where

$$\mathcal{A}_t = \left\{ \forall h,i, |\widehat{\mu}_{h,i}(t) - f(x_{h,i})| \leq \mathtt{SE}_{h,i}(T_{h,i}(t), t) \right\}. \tag{SE}$$

With a slight abuse of notation, we rewrite $\mathtt{SE}_{h,i}(T_{h,i}(t), t)$ as $\mathtt{SE}_{h,i}(T,t)$ when no confusion is caused. When $T_{h,i}(t) = 0$, $\mathtt{SE}_{h,i}(T,t)$ is naturally taken to be $+\infty$ since the node is never pulled. To ensure the above probability requirement holds, it is reasonable to make $\mathtt{SE}_{h,i}(T, t+1) \geq \mathtt{SE}_{h,i}(T,t)$ because when the number of pulls $T$ is fixed, the statistical error should not decrease.

Given the above definitions of the resolution descriptor and the uncertainty quantifier at each node, we introduce the optimum-statistical collaboration algorithm in Algorithm 1 that guides the tree-based optimum search path, under different settings of $\mathtt{OE}$ and $\mathtt{SE}$.

The basic logic behind **Algorithm 1** is that at each time $t$, the selection policy $\pi(\mathcal{S})$ will continuously search nodes from the root to leaves, until finding one node satisfying $\mathtt{OE}_{h_t} < \mathtt{SE}_{h_t,i_t}(T,t)$ and then pull this node.

The end-goal of the optimum-statistical collaboration is that, after pulling enough number of times, the following relationship holds along the shortest path from the root to the deepest node that contains the global maximum (If there are multiple global maximizers, the process only needs to find one of them) :

$$\mathtt{OE}_1 \geq \mathtt{SE}_1 > \mathtt{OE}_2 \geq \mathtt{SE}_2 \geq \cdots \geq \mathtt{OE}_h \geq \mathtt{SE}_h \geq \cdots \tag{2}$$

with slightly abused notation of $\mathtt{SE}_h$ to represent the uncertainty quantifier of the $h$-th node in the traverse path (refer to Figure 2). In other words, the two terms collaborate on the optimization process so that $\mathtt{SE}$ is controlled by $\mathtt{OE}$ for each node of the traverse path, and they both become smaller when the exploration algorithm goes deeper. Figure 2 illustrate the above dynamic process more clearly with an

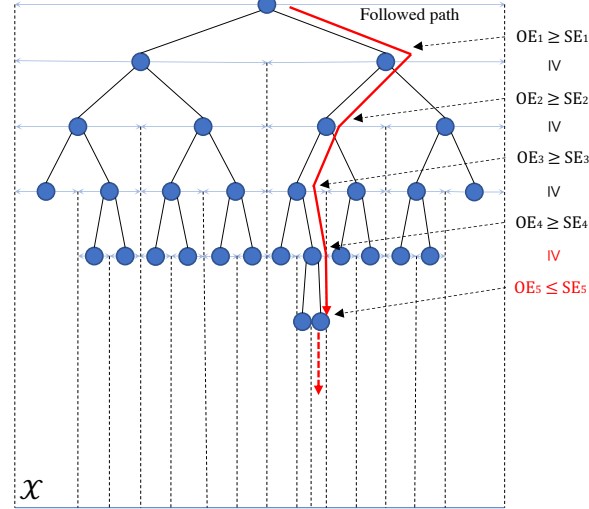

Figure 2: Illustration of the optimum-statistical collaboration framework. The node on the fifth level in the path will be pulled because its $\mathtt{OE} \leq \mathtt{SE}$

example tree on the standard partition. We remark that Eqn. (2) only needs to be guaranteed on the traverse path at each time instead of the whole exploration tree to avoid any waste of the budget. For the same purpose, all the proposed algorithms only require $\texttt{OE}_h$ to be slightly larger than or equal to $\texttt{SE}_h$ on each level.

We state the following theorem, which is a general regret upper bound with respect to any choice of $\texttt{SE}_{h,i}(T,t)$ and $\texttt{OE}_h$, and any design of the selection policy that follows the optimum statistical collaboration framework, with only a mild condition on the result of the policy in each round.

**Theorem 3.1. (General Regret Bound)** *Suppose that under a sequence of probability events $\{\mathcal{E}_t\}_{t=1,2,\cdots}$, the policy $\pi(\mathcal{S})$ ensures that at each time $t$, the node $\mathcal{P}_{h_t,i_t}$ pulled in line 9 in Algorithm 1 satisfies $f^* - f(x_{h_t,i_t}) \leq a \cdot \max\{\texttt{SE}_{h_t,i_t}(T,t),\ \texttt{OE}_{h_t}\}$, where $a > 0$ is an absolute constant. Then for any integer $\overline{H} \in [1, H(n))$ and any $0 < \delta < 1$, we have the following bound on the cumulative regret with probability at least $1 - \delta/(4n^2)$,*

$$
R_n \leq \sum_{t=1}^{n} \mathbb{I}(\mathcal{E}_t^c) + \sqrt{2n \log\left(\frac{4n^2}{\delta}\right)} + 2aC \sum_{h=1}^{\overline{H}} (\texttt{OE}_{h-1})^{-\bar{d}} \sum_{t=1}^{n} \max_{i:T_{h,i}(t) \neq 0} \texttt{SE}_{h,i}(T,t)
$$
$$
+ a \sum_{\overline{H}+1}^{H(n)} \sum_{i:T_{h,i}(t) \neq 0} \sum_{t=1}^{n} \texttt{SE}_{h,i}(T,t)
$$

*where $\bar{d} > d(a, C, \texttt{OE}_{h-1})$, $d(a, C, \texttt{OE}_{h-1})$ is the near-optimality dimension w.r.t. $a, C$, and $\texttt{OE}_{h-1}$.*

Notice that in Theorem 3.1, we do not specify the form of $\texttt{OE}_h$, $\texttt{SE}_{h,i}(T,t)$, or the specific selection policy of the algorithm. Therefore, our result is general and it can be applied to any function and partition that has a well defined $d(a, C, \texttt{OE}_{h-1})$ with resolution $\texttt{OE}_h$, and any algorithm that satisfies the algorithmic framework. The requirement $f^* - f(x_{h_t,i_t}) \leq a \cdot \max\{\texttt{SE}_{h_t,i_t}(T,t),\ \texttt{OE}_{h_t}\}$ is mild and natural in the sense that it indicates the $\pi(\mathcal{S})$ selects a "good" node to pull at each time $t$ that is at least close to the optimum relatively with respect to $\texttt{OE}$ or $\texttt{SE}$, with probability $\mathbb{P}(\mathcal{E}_t)$. Note that with a good choice of $\pi$, $\mathcal{E}_t^c$ can reduce to a subset of $\mathcal{A}_t^c$, hence $\sum_{t=1}^{n} \mathbb{I}(\mathcal{E}_t^c)$ is bounded in $L_1$. The terms that involve $\texttt{SE}$ and $\texttt{OE}$ are random due to $H(n)$, but can still be explicitly bounded when the they are well designed. Specific regret bounds for different choices of $\texttt{OE}$ and a new $\texttt{SE}$ are provided in the next section.

## 4 Implementation of Optimum-statistical Collaboration

Provided the optimum-statistical collaboration framework and its analysis, we discuss the some specific forms of the resolution descriptor and the uncertainty quantifier and elaborate the roles these definitions played in the optimization process. We then introduce a novel VHCT algorithm based on one variance-adaptive choice of $\texttt{SE}$, which is a better quantifier of the statistical uncertainty.

### 4.1 The Resolution Descriptor (Definition 1)

The resolution descriptor $\texttt{OE}$ is often measured by the global or local smoothness of the objective function (Azar et al., 2014; Grill et al., 2015). We first discuss the local smoothness assumption used by prior works and show its limitations, and then introduce a generalized local smoothness condition.

**Local Smoothness**. Grill et al. (2015) assumed that there exist two constants $\nu_1 > 0, \rho \in (0, 1)$ s.t.

$$
\forall h \geq 0, \forall x \in \mathcal{P}_{h,i_h^*}, f(x) \geq f^* - \nu_1 \rho^h. \tag{3}
$$

The above equation states that the function $f$ is $\nu_1 \rho^h$-smooth around the maximum at each level $h$. It has been considered in many prior works such as Shang et al. (2019); Bartlett et al. (2019). The resolution descriptor is naturally taken to be $\texttt{OE}_h = \nu_1 \rho^h$.

However, such a choice of local smoothness is too restrictive as it requires that the function $f$ and the partition $\mathcal{P}$ are both "well-behaved" so that the function value becomes exponentially closer to the optimum

when as $h$ increases. A simple counter-example is the function $g(x) = 1 + 1/(\ln x)$ defined on the domain $[0, 1/e]$ with $g(0)$ defined to be 0 (as shown in Figure 1(b)). Under the standard binary partition, it is easily to prove that it doesn't satisfy Eqn. (3) for any given constants $\nu_0 > 0, \rho_0 \in (0, 1)$. It might be possible to design a particular partition for $g(x)$ such that Eqn. (3) holds. However, such a partition is defined in hindsight since one have no knowledge of the objective function before the optimization. Beyond the above example, it is also easy to design other non-monotone difficult-to-optimize functions that cannot be analyzed by prior works. It thus inspires us to introduce a more general $\phi(h)$-local smoothness condition for the objective to analyze functions and partitions that have different levels of local smoothness.

**General Local Smoothness**. Assume that there exists a function $\phi(h) : \mathbb{N} \to (0, 1]$ s.t.

$$\forall h \geq 0, \forall x \in \mathcal{P}_{h,i_h^*}, f(x) \geq f(x^*) - \phi(h) \tag{GLS}$$

In the same example $g(x) = 1 + 1/(\ln x)$, it can be shown that $g(x)$ satisfies Condition (GLS) with $\phi(h) = 2/h$. Therefore, it fits in our framework by setting $\mathtt{OE}_h = 2/h$ and a valid regret bound can be obtained for $g(x)$ given a properly chosen $\mathtt{SE}_{h,i}$, since $d(2, C, 1/h) < \infty$ in this case (refer to details in Subsection 4.4). In general, we can simply set $\mathtt{OE}_h = \phi(h)$ within the optimum-statistical collaboration framework, and Theorem 3.1 can be utilized to analyze functions and partitions that satisfy Condition (GLS) with any $\phi(h)$ such as $\phi(h) = 1/h^p$, for some $p > 0$ or even $\phi(h) = 1/(\log h + 1)$, as long as the corresponding near-optimality dimension $d(a, C, \phi(h))$ is finite for some $a, C > 0$. Determining the class of smoothness functions $\phi(h)$ that can generate nontrivial regret bounds would be an interesting future direction. Given such a generalized definition and the general bound in Theorem 3.1, we can provide the convergence analysis for a much larger class of black-box objectives and partitions, beyond those that satisfy Eqn. (3).

## 4.2 The Uncertainty Quantifier (Definition 2)

**Tracking Statistics**. To facilitate the design of $\mathtt{SE}$, we first define the following tracking statistics. Trivially, the *mean estimate* $\widehat{\mu}_{h,i}(t)$ and the *variance estimate* $\widehat{\mathbb{V}}_{h,i}(t)$ of the rewards at round $t$ are computed as

$$\widehat{\mu}_{h,i}(t) \equiv \frac{1}{T_{h,i}(t)} \sum_{k=1}^{T_{h,i}(t)} r^k(x_{h,i}), \quad \widehat{\mathbb{V}}_{h,i}(t) \equiv \frac{1}{T_{h,i}(t)} \sum_{k=1}^{T_{h,i}(t)} \left( r^k(x_{h,i}) - \widehat{\mu}_{h,i}(t) \right)^2$$

The variance estimate is defined to be negative infinity when $T_{h,i}(t) = 0$ since variance is undefined in such cases. We now discuss two specific choices of $\mathtt{SE}$.

**Nonadaptive Quantifier** (in $\mathtt{HCT}$). Azar et al. (2014) proposed the uncertainty quantifier with the following form in their High Confidence Tree ($\mathtt{HCT}$) algorithm:

$$\mathtt{SE}_{h,i}(T, t) \equiv bc\sqrt{\frac{\log(\Delta(t))}{T_{h,i}(t)}}$$

where $b/2$ is the bound of the error noise $\epsilon_t$, $\Delta(t) = \max\{1, 2^{\lfloor \log t \rfloor + 1}/(c_1\delta)\}$ is an increasing function of $t$, $\delta$ is the confidence level, and $c, c_1$ are two tuning constants. By Hoeffding's inequality, the above $\mathtt{SE}$ is a high-probability upper bound for the statistical uncertainty. Note that $\mathtt{HCT}$ is also a special case of our OSC framework and its analysis can be done by following Theorem 3.1. In what follows, we propose an better algorithm with an improved uncertainty quantifier.

**Variance Adaptive Quantifier** (in $\mathtt{VHCT}$). Based on our framework of the statistical collaboration, a tighter measure of the statistical uncertainty can boost the performance of the optimization algorithm, as the goal in Eqn. (2) can be reached faster. Motivated by prior works that use variance to improve the performance of multi-armed bandit algorithms Audibert et al. (2006; 2009), we propose the following variance adaptive uncertainty quantifier, and naturally the $\mathtt{VHCT}$ algorithm in the next subsection, which is an adaptive variant of the $\mathtt{SE}$ in $\mathtt{HCT}$.

$$\mathtt{SE}_{h,i}(T, t) \equiv c\sqrt{\frac{2\widehat{\mathbb{V}}_{h,i}(t)\log(\Delta(t))}{T_{h,i}(t)}} + \frac{3bc^2\log(\Delta(t))}{T_{h,i}(t)} \tag{4}$$

---

**Algorithm 2** `VHCT` Algorithm (Short Version)

---

1: **Input**: known smoothness function $\phi(h)$, partition $\mathcal{P}$.
2: Run Algorithm 1 with partition $\mathcal{P}$ and other required inputs as:

$$\mathtt{OE}_h := \phi(h), \ \mathtt{SE}_{h,i}(T,t) := \ \text{Eqn. (4)}, \ \pi(\mathcal{S}) := \mathrm{argmax}_{\mathcal{P}_{h,i} \in \mathcal{S}} B_{h,i}(t)$$

---

The notations $b, c$, and $\Delta(t)$ are the same as those in `HCT`. The uniqueness of the above $\mathtt{SE}_{h,i}(T,t)$ is that it utilizes the node-specified variance estimations, instead of a conservative trivial bound $b$. Therefore, the algorithm is able to adapt to different noises across nodes, and $\mathtt{SE}_{h,i}(T,t) \leq \mathtt{OE}_h$ is achieved faster at the small-noise nodes. This unique property grants `VHCT` an advantage over all existing non-adaptive algorithms.

### 4.3 Algorithm Example - `VHCT`

Based on the proposed optimum-statistical collaboration framework and the novel adaptive $\mathtt{SE}_{h,i}(T,t)$, we propose a new algorithm `VHCT` as a special case of Algorithm 1 and elaborate its capability to adapt to different noises. Algorithm 2 provides the short version of the pseudo-code and the complete algorithm is provided in Appendix B.

The proposed `VHCT`, similar to `HCT`, also maintains an upper-bound $U_{h,i}(t)$ for each node to decide collaborative optimism. In particular, for any node $\mathcal{P}_{h,i}$, the upper-bound $U_{h,i}(t)$ is computed directly from the average observed reward for pulling $x_{h,i}$ as

$$U_{h,i}(t) \equiv \widehat{\mu}_{h,i}(t) + \mathtt{OE}_h + \mathtt{SE}_{h,i}(T,t)$$

with $\mathtt{SE}_{h,i}(T,t)$ defined as in Eqn. (4) and $\mathtt{OE}_h$ tuned by the input. Note that $U_{h,i}(t) = \infty$ for unvisited nodes. To better utilize the tree structure in the algorithm, we also define the tighter upper bounds $B_{h,i}(t)$. Since the maximum upper bound of one node cannot be greater than the maximum of its children, $B_{h,i}(t)$ is defined to be

$$B_{h,i}(t) = \min \left\{ U_{h,i}(t), \max_{j=0,1}\{B_{h+1,2i-j}(t)\} \right\}.$$

The quantities $U_{h,i}(t)$ and $B_{h,i}(t)$ serve a similar role of the upper confidence bound in UCB bandit algorithm (Bubeck et al., 2011b), and the selection policy $\pi(\mathcal{S})$ of `VHCT` is simply selecting the node with the highest $B_{h,i}(t)$ in the given set $\mathcal{S}$, which is shown in Algorithm 2. We prove that selection policy guarantees that $f^* - f(x_{h_t,i_t}) \leq 3\max\{\mathtt{SE}_{h_t,i_t}(T,t), \mathtt{OE}_{h_t}\}$ with high probability in Appendix B, as we required in Theorem 3.1.

Follow the notation of Azar et al. (2014), we define a threshold value $\tau_{h,i}(t)$ for each node $\mathcal{P}_{h,i}$ to represent the minimal number of times it has been pulled, such that the algorithm can explore its children nodes, i.e.,

$$\tau_{h,i}(t) = \inf_{T \in \mathbb{N}} \left\{ \mathtt{SE}_{h,i}(T,t) \leq \mathtt{OE}_h \right\}.$$

Only when $T_{h_t,i_t}(t) \geq \tau_{h_t,i_t}(t)$, we expand the search into $\mathcal{P}_{h_t,i_t}$'s children. This notation helps to compare the exploration power of `VHCT` with `HCT`. Note that when the variances of the nodes are small, $\mathtt{SE}_{h,i}(T,t)$ of `VHCT` would be inversely proportional to $T_{h,i}(t)$ and thus smaller than that of `HCT`. As a consequence, the thresholds $\tau_{h,i}(t)$ is smaller in `VHCT` than in `HCT`, and thus `VHCT` explores more efficiently in low noise regimes.

### 4.4 Regret Bound Examples

We now provide upper bounds on the expected cumulative regret of `VHCT`, which serve as instances of our general Theorem 3.1 when `OE` and `SE` are specified. Note that some technical adaptions are made to obtain a $L_1$ bound for the regret. The regret bounds depend on the upper bound of variance in history across all the nodes that have been pulled, meaning $\max_{\{h,i,t|T_{h,i}(t)\geq 1\}} \widehat{\mathbb{V}}_{h,i}(t) \leq V_{\max}$ for a constant $V_{\max} > 0$. Since

the noise $\epsilon_t$ is bounded, such a notation is always well defined and bounded above. The $V_{\max}$ represents our knowledge of the noise variance after searching and exploring the objective function, which can be more accurate than the trivial choice $b^2/4$, e.g., when the true noise is actually bounded by $b'/2$ for some unknown constant $b' < b$. We focus on two choices of the local smoothness function in Condition (GLS) and their corresponding near-optimal dimensions, i.e., $\phi(h) = \nu_1 \rho^h$ that matches previous analyses such as Grill et al. (2015); Shang et al. (2019), and $\phi(h) = 2/h$, which is the local smoothness of the counter example in Figure 1(b). For other choices of $\phi(h)$, we believe similar regret upper bounds may be derived using Theorem 3.1.

**Theorem 4.1.** *Assume that the objective function $f$ satisfies Condition* (GLS) *with $\phi(h) = \nu_1 \rho^h$ for two constants $\nu_1 > 0, \rho \in (0,1)$. The expected cumulative regret of Algorithm 3 is upper bounded by*

$$\mathbb{E}[R_n^{\mathtt{VHCT}}] \leq 2\sqrt{2n\log(4n^3)} + C_1 V_{\max}^{\frac{1}{d_1+2}} n^{\frac{d_1+1}{d_1+2}} (\log n)^{\frac{1}{d_1+2}} + C_2 n^{\frac{2d_1+1}{2d_1+4}} \log n$$

*where $C_1$ and $C_2$ are two constants and $d_1$ is any constant satisfying $d_1 > d(3\nu_1, C, \rho^h)$.*

**Theorem 4.2.** *Assume that the objective function $f$ satisfies Condition* (GLS) *with $\phi(h) = 2/h$. The expected cumulative regret of Algorithm 3 is upper bounded by*

$$\mathbb{E}[R_n^{\mathtt{VHCT}}] \leq 2\sqrt{2n\log(4n^3)} + \bar{C}_1 V_{\max}^{\frac{1}{2d_2+3}} n^{\frac{2d_2+2}{2d_2+3}} (\log n)^{\frac{1}{2d_2+3}} + \bar{C}_2 n^{\frac{2d_2+1}{2d_2+3}} \log n$$

*where $\bar{C}_1$ and $\bar{C}_2$ are two constants and $d_2$ is any constant satisfying $d_2 > d(2, C, 1/h)$.*

The proof of these theorems are provided in Appendix C and Appendix D respectively. We first remark that the above regret bounds are actually loose because we do not a delicate individual control over the variances in different nodes. Instead, a much conservative analysis is conducted.

In the literature, Grill et al. (2015); Shang et al. (2019) have proved that the cumulative regret bounds of HOO, HCT are both $\mathcal{O}(n^{(d_1+1)/(d_1+2)}(\log n)^{1/(d_1+2)})$ when the objective function $f$ satisfies Condition (GLS) with $\phi(h) = \nu_1 \rho^h$, while our regret bound in Theorem 4.1 is of order $\mathcal{O}(V_{\max}^{1/(d_1+2)} n^{(d_1+1)/(d_1+2)}(\log n)^{1/(d_1+2)})$. Although the two rates are the same with respect to the increasing of $n$, our result explicitly connects the variance and the regret, implying a positive relationship between these two. Therefore, we expect the variance adaptive algorithm VHCT to converge faster than the non-adaptive algorithms such as HOO and HCT, when there is only low or moderate noise. The theoretical results of prior works rely on the smoothness assumption $\phi(h) = \nu_1 \rho^h$, thus are not able to deliver a regret analysis for functions and partitions with other $\phi(h)$ (e.g. $\phi(h) = 2/h$ in Theorem 4.2). Providing analysis for prior algorithms on functions and partitions with different smoothness assumptions is another interesting future direction to explore. However, we conjecture that VHCT should still outperform the non-adaptive algorithms in these cases since its SE is a tighter measure of the statistical uncertainty. This theoretical observation is also validated in our experiments.

We emphasize that the near-optimality dimensions are defined with respect to different local smoothness functions in Theorem 4.1 and Theorem 4.2. Specifically, when the objective is $\nu_1 \rho^h$-smooth, Theorem 4.1 holds even if the number of near-optimal regions increase exponentially when the partition proceeds deeper, i.e., when $d(\nu_1, C, \rho^h) < \infty$. When the function is only $2/h$-smooth, Theorem 4.2 holds only when the number of near-optimal regions grows polynomially, i.e., when $d(2, C, 1/h) < \infty$.

## 5   Experiments

In this section, we empirically compare the proposed VHCT algorithm with the existing **anytime** blackbox optimization algorithms, including T-HOO (the truncated version of HOO), HCT, POO, and PCT (POO + HCT, (Shang et al., 2019)), and Bayesian Optimization algorithm BO (Frazier, 2018) to validate that the proposed variance-adaptive uncertainty quantifier can make the convergence of VHCT faster than non-adaptive algorithms. We run every algorithm for 20 independent trials in each experiment and plot the average cumulative regret with 1-standard deviation error bounds. The experimental details and additional numerical results on other objectives are provided in Appendix E.

We use a noisy Garland function as the synthetic objective, which is a typical blackbox objective used by many works such as Shang et al. (2019) and has multiple local minimums and thus very hard to optimize.

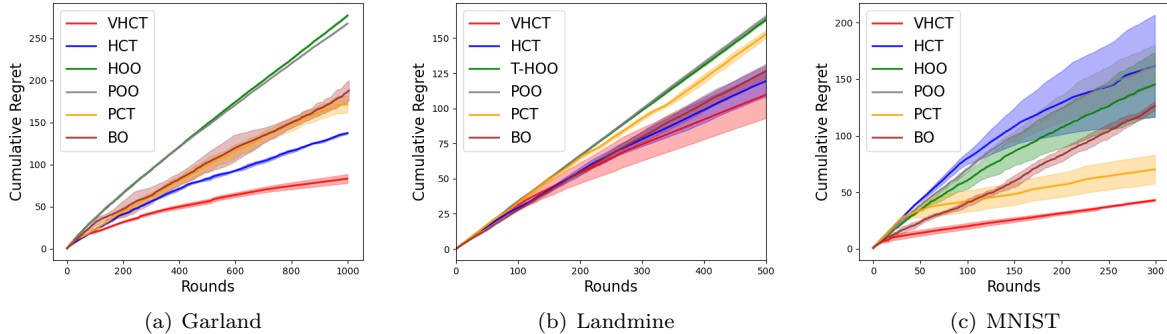

Figure 3: Cumulative regret of different algorithms on evaluating the Garland function and tuning hyperparameters of training SVM on Landmine data and neural networks on MNIST data.

For the real-life experiments, we use hyperparameter tuning of machine learning algorithms as the blackbox objectives. We tune the RBF kernel and the L2 regularization parameters when training Support Vector Machine (SVM) on the Landmine dataset (Liu et al., 2007), and the batch size, the learning rate, and the weight decay when training neural networks on the MNIST dataset (Deng, 2012). As shown in Figure 3, the new choice of `SE` makes `VHCT` the fastest algorithm among the existing ones. All the experimental results validate our theoretical claims in Section 4.

## 6 Conclusions

The proposed optimum-statistical collaboration framework reveals and utilizes the fundamental interplay of resolution and uncertainty to design more general and efficient black-box optimization algorithms. Our analysis shows that different regret guarantees can be obtained for functions and partitions with different local smoothness assumptions, and algorithms that have different uncertainty quantifiers. Based on the framework, we show that functions that satisfy the general local smoothness property can be optimized and analyzed, which is a much larger class of functions compared with prior works. Also, we propose a new algorithm `VHCT` that can adapt to different noises and analyze its performance under different assumptions of the smoothness of the function.

There are still some limitations of our work. For example, `VHCT` still needs the prior knowledge of the smoothness function $\phi(h)$ to achieve its best performance. Also, the analyses in Theorem 4.1 and 4.2 are smoothness-specific. Therefore, our framework also introduces many interesting future working directions, for example, (1) whether a unified regret upper bound for different $\phi(h)$-local smooth functions could be derived for one particular algorithm; (2) whether the regret bound obtained in Theorem 4.2 is minimax-optimal for those $\phi(h)$; (3) whether there exists an algorithm that is truly smoothness-agnostic, i.e., it does not need the smoothness property of the objective function.

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

# A   Proof of the General Regret Bound in Theorem 3.1

**Proof.**  We decompose the cumulative regret into two terms that depend on the high probability events $\{\mathcal{E}_t\}_{t=1}^n$. Denote the simple regret at each iteration $t$ to be $\Delta_t = f^* - r_t$, then we can perform the following regret decomposition

$$R_n = \sum_{t=1}^n \Delta_t = \left( \sum_{t=1}^n \Delta_t \mathbb{I}_{\mathcal{E}_t} \right) + \left( \sum_{t=1}^n \Delta_t \mathbb{I}_{\mathcal{E}_t^c} \right) = R_n^{\mathcal{E}} + R_n^{\mathcal{E}^c}$$

$$\leq R_n^{\mathcal{E}} + \sum_{t=1}^n \mathbb{I}_{\mathcal{E}_t^c}$$

where we have denoted the first summation term in the second equality $\sum_{t=1}^n \Delta_t \mathbb{I}_{\mathcal{E}_t}$ to be $R_n^{\mathcal{E}}$ and the second summation term $\sum_{t=1}^n \Delta_t \mathbb{I}_{\mathcal{E}_t^c}$ to be $R_n^{\mathcal{E}^c}$. The last inequality is because we have that both $f^*$ and $r_t$ are bounded by $[0, 1]$, and thus $|\Delta_t| \leq 1$. Now note that the instantaneous regret $\Delta_t$ can be written as

$$\Delta_t = f^* - r_t = f^* - f\left( x_{h_t, i_t} \right) + f\left( x_{h_t, i_t} \right) - r_t = \Delta_{h_t, i_t} + \widehat{\Delta}_t$$

where we have denoted $\Delta_{h_t, i_t} = f^* - f\left( x_{h_t, i_t} \right)$ and $\widehat{\Delta}_t = f\left( x_{h_t, i_t} \right) - r_t$. It means that the regret under the events $\{\mathcal{E}_t\}_{t=1}^n$ can be decomposed into two terms $\widetilde{R}_n^{\mathcal{E}}$ and $\widehat{R}_n^{\mathcal{E}}$.

$$R_n^{\mathcal{E}} = \sum_{t=1}^n \Delta_{h_t, i_t} \mathbb{I}_{\mathcal{E}_t} + \sum_{t=1}^n \widehat{\Delta}_t \mathbb{I}_{\mathcal{E}_t} \leq \sum_{t=1}^n \Delta_{h_t, i_t} \mathbb{I}_{\mathcal{E}_t} + \sum_{t=1}^n \widehat{\Delta}_t = \widetilde{R}_n^{\mathcal{E}} + \widehat{R}_n^{\mathcal{E}}$$

Note that by the definition of the sequence $\{\widehat{\Delta}_t\}_{t=1}^n$, it is a bounded martingale difference sequence since $\mathbb{E}[\widehat{\Delta}_t \mid \mathcal{F}_{t-1}] = 0$ and $|\widehat{\Delta}_t| \leq 1$, where $\mathcal{F}_t$ is defined to be the filtration generated up to time $t$. Therefore by Azuma's inequality on this sequence, we get

$$\widehat{R}_n^{\mathcal{E}} \leq \sqrt{2n \log\left( \frac{4n^2}{\delta} \right)}$$

with probability $1 - \delta/(4n^2)$. A even better bound can be obtained using the fact that $|\widehat{\Delta}_t| \leq \frac{b}{2}$ if $b \ll 2$. However, $\widehat{R}_n^{\mathcal{E}}$ is not a dominating term and using $b/2$ only improves it in terms of the multiplicative constant. Now the only term left is $\widetilde{R}_n^{\mathcal{E}}$ and we bound it as follows.

$$\widetilde{R}_n^{\mathcal{E}} = \left( \sum_{t=1}^n \Delta_{h_t, i_t} \mathbb{I}_{\mathcal{E}_t} \right) \leq \left( \sum_{h=1}^{H(n)} \sum_{i:T_{h,i}(t) \neq 0} \sum_{t=1}^n \Delta_{h,i} \mathbb{I}_{(h_t, i_t) = (h, i)} \mathbb{I}_{\mathcal{E}_t} \right)$$

$$\leq \sum_{h=1}^{\overline{H}} \sum_{i:T_{h,i}(t) \neq 0} \sum_{t=1}^n a\mathrm{SE}_{h,i}(T, t) + \sum_{\overline{H}+1}^{H(n)} \sum_{i:T_{h,i}(t) \neq 0} \sum_{t=1}^n a\mathrm{SE}_{h,i}(T, t)$$

$$\leq \underbrace{a \sum_{h=1}^{\overline{H}} \sum_{i:T_{h,i}(t) \neq 0} \sum_{t=1}^n \mathrm{SE}_{h,i}(T, t)}_{\text{(I)}} + \underbrace{a \sum_{\overline{H}+1}^{H(n)} \sum_{i:T_{h,i}(t) \neq 0} \sum_{t=1}^n \mathrm{SE}_{h,i}(T, t)}_{\text{(II)}}$$

where $\overline{H}$ is a constant between 0 and $H(n)$ to be tuned later. The second inequality is because when we select $\mathcal{P}_{h_t, i_t}$, we have $\mathrm{SE}_{h_t, i_t}(T, t) \geq \mathrm{OE}_{h_t}$ by the Optimum-statistical Collaboration Framework. Also, under

the event $\mathcal{E}_t$, we have $\Delta_{h_t,i_t} \leq a \max\{\texttt{OE}_{h_t}, \texttt{SE}_{h_t,i_t}(T,t)\}$. The first term (I) can be bounded as

$$(\text{I}) \leq a \sum_{h=1}^{\overline{H}} \sum_{i:T_{h,i}(t)\neq 0} \sum_{t=1}^{n} \max_{i:T_{h,i}(t)\neq 0} \texttt{SE}_{h,i}(T,t) \leq a \sum_{h=1}^{\overline{H}} |\mathcal{I}_h(n)| \sum_{t=1}^{n} \max_{i:T_{h,i}(t)\neq 0} \texttt{SE}_{h,i}(T,t)$$

$$\leq a \sum_{h=1}^{\overline{H}} 2\mathcal{N}_{h-1}\left(a\texttt{OE}_{h-1}\right) \sum_{t=1}^{n} \max_{i:T_{h,i}(t)\neq 0} \texttt{SE}_{h,i}(T,t)$$

$$\leq 2aC \sum_{h=1}^{\overline{H}} \left(\texttt{OE}_{h-1}\right)^{-\bar{d}} \sum_{t=1}^{n} \max_{i:T_{h,i}(t)\neq 0} \texttt{SE}_{h,i}(T,t)$$

where $\bar{d} > d(a,C,\texttt{OE}_{h-1})$ and $d(a,C,\texttt{OE}_{h-1})$ is the near-optimality dimension with respect to $(a,C,\texttt{OE}_{h-1})$. The third inequality is because we only expand a node into two children, so $|\mathcal{I}_h(n)| \leq 2|\mathcal{I}_{h-1}^+(n)|$ (Note that we do not have any requirements on the number of children of each node, so the binary tree argument here can be easily replaced by a $K$-nary tree with $K \geq 2$. Also since we only select a node $(h,i)$ when its parent is already selected enough number of times such that $\texttt{OE} \geq \texttt{SE}$ at a particular time $t_0 \leq n$, we have $\mathcal{P}_{h^p,i^p}$ satisfies $f^* - f(x_{h^p,i^p}) \leq a\texttt{OE}_{h^p}$. By the definition of $\mathcal{N}_h(\epsilon)$ in the near-optimality dimension, we have

$$|\mathcal{I}_h(n)| \leq 2|\mathcal{I}_{h-1}^+(n)| \leq 2\mathcal{N}_{h-1}\left(a\texttt{OE}_{h-1}\right)$$

and thus the final upper bound for (I). Therefore for any $\overline{H} \in [1, H(n)]$, with probability at least $1 - \frac{\delta}{4n^2}$, the cumulative regret is upper bounded by

$$R_n = \sum_{t=1}^{n} \Delta_t = \widehat{R}_n^{\mathcal{E}} + \sum_{t=1}^{n} \mathbb{I}(\mathcal{E}_t^c) + \widetilde{R}_n^{\mathcal{E}}$$

$$\leq \sqrt{2n\log(4n^2/\delta)} + \sum_{t=1}^{n} \mathbb{I}(\mathcal{E}_t^c) + \widetilde{R}_n^{\mathcal{E}}$$

$$\leq \sqrt{2n\log(4n^2/\delta)} + \sum_{t=1}^{n} \mathbb{I}(\mathcal{E}_t^c) + 2aC \sum_{h=1}^{\overline{H}} \left(\texttt{OE}_{h-1}\right)^{-\bar{d}} \sum_{t=1}^{n} \max_{i:T_{h,i}(t)\neq 0} \texttt{SE}_{h,i}(T,t)$$

$$+ a \sum_{\overline{H}+1}^{H(n)} \sum_{i:T_{h,i}(t)\neq 0} \sum_{t=1}^{n} \texttt{SE}_{h,i}(T,t)$$

$$\square$$

## B  Notations and Useful Lemmas

### B.1  Preliminary Notations

The notations here follow those in Shang et al. (2019) and Azar et al. (2014) except for those related to the node variance. These notations are needed for the proof of the main theorem.

- At each time $t$, $\mathcal{P}_{h_t,i_t}$ denote the node selected by the algorithm where $h_t$ is the level and $i_t$ is the index.

- $P_t$ denotes the optimal-path selected at each iteration $t$

- $H(t)$ denotes the maximum depth of the tree at time $t$.

- $\Delta(t) = 1/\tilde{\delta}(t^+)$ with $t^+ = 2^{\lfloor \log t \rfloor + 1}$, $\tilde{\delta}(t) = \min\{1, c_1\delta/t\}$

- For any $t > 0$ and $h \in [1, H(t)]$, $\mathcal{I}_h(t)$ denotes the set of all nodes at level $h$ at time $t$.

- For any $t > 0$ and $h \in [1, H(t)]$, $\mathcal{I}_h^+(t)$ denotes the subset of $\mathcal{I}_h(t)$ that contains only the internal nodes (no leaves).

- $\mathcal{C}_{h,i} := \{t \in [1, n] \mid \mathcal{P}_{h_t, i_t} = \mathcal{P}_{h,i}\}$ is the set of time steps when $\mathcal{P}_{h,i}$ is selected.

- $\mathcal{C}_{h,i}^+ := \mathcal{C}_{h+1,2i} \cup \mathcal{C}_{h+1,2i-1}$ is the set of time steps when the children of $\mathcal{P}_{h,i}$ are selected.

- $\bar{t}_{h,i} := \max_{t \in \mathcal{C}_{h,i}} t$ is the last time $\mathcal{P}_{h,i}$ is selected.

- $\tilde{t}_{h,i} := \max_{t \in \mathcal{C}_{h,i}^+} t$ is the last time when the children of $\mathcal{P}_{h,i}$ is selected.

- $t_{h,i} := \min \{t : T_{h,i}(t) \geq \tau_{h,i}\}$ is the time when $\mathcal{P}_{h,i}$ is expanded.

- $\widehat{\mathbb{V}}_{h,i}(t) := \frac{1}{T_{h,i}(t)} \sum_{s=1}^{T_{h,i}(t)} \left(r^s(x_{h,i}) - \widehat{\mu}_{h,i}\right)$ is the estimate of the variance of the $\mathcal{P}_{h,i}$ node at time $t$.

- $\mathcal{L}_t$ denotes all the nodes in the exploration tree at time $t$

- $V_{\max}$ is the upper bound on the node variance in the tree.

Note that if the variance of a node is zero, we can always pull one more round to make it non-zero. Therefore, here we simply assume that the variance $\mathbb{V}_{h,i}(t)$ is larger than a fixed small constant $\epsilon$ for the clarity of proof, which will not affect our conclusions.

---

**Algorithm 3** `VHCT` Algorithm (Complete)

---

1: **Input:** Smoothness function $\phi(h)$, partition $\mathcal{P}$.
2: **Initialize:** $\mathcal{T}_t = \{\mathcal{P}_{0,1}, \mathcal{P}_{1,1}, \mathcal{P}_{1,2}\}, U_{1,1}(t) = U_{1,2}(t) = +\infty$
3: **for** $t = 1$ to $n$ **do**
4:   **if** $t = t^+$ **then**
5:     **for** all nodes $\mathcal{P}_{h,i} \in \mathcal{T}_t$ **do**
6:       $U_{h,i}(t) = \mu_{h,i}(t) + \phi(h) + \text{SE}_{h,i}(T, t)$
7:     **end for**
8:     $\text{UpdateBackward}(\mathcal{T}_t, t)$
9:   **end if**
10:   $\mathcal{P}_{h_t, i_t} = \text{PullUpdate}(\mathcal{T}_t, t)$
11:   **if** $T_{h_t, i_t}(t) \geq \tau_{h_t, i_t}(t)$ and $\mathcal{P}_{h_t, i_t}$ is a leaf **then**
12:     $\mathcal{T}_t = \mathcal{T}_t \cup \{\mathcal{P}_{h_t+1, 2i_t-1}, \mathcal{P}_{h_t+1, 2i_t}\}$
13:     $U_{h+1, 2i}(t) = U_{h+1, 2i-1}(t) = +\infty$
14:   **end if**
15: **end for**

---

### B.2 Useful Lemmas for the Proof of Theorem 4.1 and Theorem 4.2

The following lemma improves the results by Azar et al. (2014) and Shang et al. (2019).

**Lemma B.1.** *We introduce the following event $\mathcal{E}_t$*

$$
\mathcal{E}_t = \left\{ \forall \mathcal{P}_{h,i} \in \mathcal{L}_t, \forall T_{h,i}(t) = 1, \cdots, t : |\widehat{\mu}_{h,i}(t) - f(x_{h,i})| \leq c\sqrt{\frac{2\widehat{\mathbb{V}}_{h,i}(t) \log(1/\tilde{\delta}(t))}{T_{h,i}(t)}} + \frac{3bc^2 \log(1/\tilde{\delta}(t))}{T_{h,i}(t)} \right\}
$$

*where $x_{h,i} \in \mathcal{P}_{h,i}$ is the arm corresponding to node $\mathcal{P}_{h,i}$. If*

$$
c = 3 \quad and \quad \tilde{\delta}(t) = \frac{\delta}{3t}
$$

*then for any fixed $t$, the event $\mathcal{E}_t$ holds with probability at least $1 - \delta/t^7$.*

---

**Algorithm 4** `PullUpdate`

---

1: **Input:** a tree $\mathcal{T}_t$, round $t$
2: **Initialize:** $(h_t, i_t) = (0, 1); S_t = \mathcal{P}_{0,1}; T_{0,1}(t) = \tau_0(t) = 1$ ;
3: **while** $\mathcal{P}_{h_t, i_t}$ is not a leaf, $T_{h_t, i_t}(t) \geq \tau_{h_t, i_t}(t)$ **do**
4:     $j = \text{argmax}_{j=0,1}\{B_{h_t+1, 2i_t-j}(t)\}$
5:     $(h_t, i_t) = (h_t + 1, 2i_t - j)$
6:     $S_t = S_t \cup \{\mathcal{P}_{h_t, i_t}\}$
7: **end while**
8: Pull $x_{h_t, i_t}$ and get reward $r_t$
9: $T_{h_t, i_t}(t) = T_{h_t, i_t}(t) + 1$
10: Update $\widehat{\mu}_{h_t, i_t}(t), \widehat{\mathbb{V}}_{h_t, i_t}(t)$
11: $U_{h_t, i_t}(t) = \widehat{\mu}_{h_t, i_t}(t) + \phi(h_t) + \text{SE}_{h_t, i_t}(T, t)$
12: `UpdateBackward`$(S_t, t)$
13: **Return** $\mathcal{P}_{h_t, i_t}$

---

**Algorithm 5** `UpdateBackward`

---

1: **Input:** a tree $\mathcal{T}$, round $t$
2: **for** $\mathcal{P}_{h,i} \in \mathcal{T}$ backward from each leaf of $\mathcal{T}$ **do**
3:     **if** $\mathcal{P}_{h,i}$ is a leaf of $\mathcal{T}$ **then**
4:         $B_{h,i}(t) = U_{h,i}(t)$
5:     **else**
6:         $B_{h,i}(t) = \min\{U_{h,i}(t), \max_j\{B_{h+1, 2i-j}(t)\}\}$
7:     **end if**
8:     Update the threshold $\tau_{h,i}(t)$
9: **end for**

---

**Proof.** Again, $\mathcal{L}_t$ denotes all the nodes in the tree. The probability of $\mathcal{E}_t^c$ can be bounded as

$$\mathbb{P}\left[\mathcal{E}_t^c\right] \leq \sum_{\mathcal{P}_{h,i} \in \mathcal{L}_t} \sum_{T_{h,i}(t)=1}^{t} \mathbb{P}\left[|\widehat{\mu}_{h,i}(t) - \mu_{h,i}| \geq c\sqrt{\frac{2\widehat{\mathbb{V}}_{h,i}(t)\log(1/\tilde{\delta}(t))}{T_{h,i}(t)}} + \frac{3bc^2\log(1/\tilde{\delta}(t))}{T_{h,i}(t)}\right]$$

$$\leq \sum_{\mathcal{P}_{h,i} \in \mathcal{L}_t} \sum_{T_{h,i}(t)=1}^{t} 3\exp(-c^2\log(1/\tilde{\delta}(t)))$$

$$= 3\exp(-c^2\log(1/\tilde{\delta}(t))) \cdot t \cdot |\mathcal{L}_t|$$

where the second inequality is by taking $x = c^2\log(1/\tilde{\delta}(t))$ in Lemma B.6, we have

$$\mathbb{P}\left(|\widehat{\mu}_{h,i}(t) - f(x_{h,i})| \geq c\sqrt{\frac{2\widehat{\mathbb{V}}_{h,i}(t)\log(1/\tilde{\delta}(t))}{T_{h,i}(t)}} + \frac{3bc^2\log(1/\tilde{\delta}(t))}{T_{h,i}(t)}\right) \leq 3\exp(-c^2\log(1/\tilde{\delta}(t)))$$

Now note that the number of nodes in the tree is always (loosely) bounded by $t$ since we need at least one pull to expand a node, we know that

$$\mathbb{P}\left[\mathcal{E}_t^c\right] \leq 3t^2\tilde{\delta}(t)^{c^2} \leq \frac{\delta}{t^7} \qquad \square$$

**Lemma B.2.** *Given the parameters $c$ and $\tilde{\delta}(t)$ as in Lemma B.1, the regret when the events $\{\mathcal{E}_t\}$ fail to hold is bounded as*

$$\sum_{t=1}^{n} \mathbb{I}(\mathcal{E}_t^c) \leq \sqrt{n}$$

*with probability at least $1 - \delta/(6n^3)$*

**Proof.** We first split the time horizon $n$ in two phases: the first phase until $\sqrt{n}$ and the rest. Thus the regret bound becomes

$$\sum_{t=1}^{n} \mathbb{I}(\mathcal{E}_t^c) = \sum_{t=1}^{\sqrt{n}} \mathbb{I}(\mathcal{E}_t^c) + \sum_{t=\sqrt{n}+1}^{n} \mathbb{I}(\mathcal{E}_t^c)$$

The first term can be easily bounded by $\sqrt{n}$. Now we bound the second term by showing that the complement of the high-probability event hardly ever happens after $t = \sqrt{n}$. By Lemma B.1

$$\mathbb{P}\left[\bigcup_{t=\sqrt{n}+1}^{n} \mathcal{E}_t^c\right] \leq \sum_{t=\sqrt{n}+1}^{n} \mathbb{P}\left[\mathcal{E}_t^c\right] \leq \sum_{\sqrt{n}+1}^{n} \frac{\delta}{t^7} \leq \int_{\sqrt{n}}^{+\infty} \frac{\delta}{t^7} dt \leq \frac{\delta}{6n^3}$$

Therefore we arrive to the conclusion in the lemma. $\qquad\square$

**Lemma B.3.** *At time $t$ under the event $\mathcal{E}_t$, for the selected node $\mathcal{P}_{h_t,i_t}$ and its parent $(h_t^p, i_t^p)$, we have the following set of inequalities for any choice of the local smoothness function $\phi(h)$ in Algorithm 3*

$$\begin{cases} f^* - f(x_{h_t,i_t}) \leq 3c\sqrt{\dfrac{2\widehat{\mathbb{V}}_{h_t,i_t}(t)\log(2/\tilde{\delta}(t))}{T_{h_t,i_t}(t)}} + \dfrac{9bc^2\log(2/\tilde{\delta}(t))}{T_{h_t,i_t}(t)} \\[4mm] f^* - f(x_{h_t^p,i_t^p}) \leq 3\phi(h_t^p) \end{cases}$$

**Proof.** Recall that $P_t$ is the optimal path traversed. Let $(h', i') \in P_t$ and $(h'', i'')$ be the node which immediately follows $(h', i')$ in $P_t$ (i.e., $h'' = h' + 1$). By the definition of $B$ values, we have the following inequality

$$B_{h',i'}(t) \leq \max\left(B_{h'+1,2i'-1}(t); B_{h'+1,2i'}(t)\right) = B_{h'',i''}(t)$$

where the last equality is from the fact that the algorithm selects the child with the larger $B$ value. By iterating along the inequality until the selected node $(h_t, i_t)$ and its parent $(h_t^p, i_t^p)$ we obtain

$$\forall (h', i') \in P_t, \quad B_{h',i'}(t) \leq B_{h_t,i_t}(t) \leq U_{h_t,i_t}(t),$$
$$\forall (h', i') \in P_t - (h_t, i_t), \quad B_{h',i'}(t) \leq B_{h_t^p,i_t^p}(t) \leq U_{h_t^p,i_t^p}(t),$$

Thus for any node $\mathcal{P}_{h,i} \in P_t$, we have that $U_{h_t,i_t}(t) \geq B_{h,i}(t)$. Furthermore, since the root node $(0,1)$ is a an optimal node in the path $P_t$. Therefore, there exists at least one node $(h^*, i^*) \in P_t$ which includes the maximizer $x^*$ and has the the the depth $h^* \leq h_t^p < h_t$. Thus

$$U_{h_t,i_t}(t) \geq B_{h^*,i^*}(t), \quad U_{h_t^p,i_t^p}(t) \geq B_{h^*,i^*}(t)$$

Note that by the definition of $U_{h_t,i_t}(t)$, under event $\mathcal{E}_t$

$$\begin{aligned}
U_{h_t,i_t}(t) &= \widehat{\mu}_{h_t,i_t}(t) + \phi(h_t) + c\sqrt{\frac{2\widehat{\mathbb{V}}_{h_t,i_t}(t)\log(1/\tilde{\delta}(t^+))}{T_{h_t,i_t}(t)}} + \frac{3bc^2\log(1/\tilde{\delta}(t^+))}{T_{h_t,i_t}(t)} \\[2mm]
&\leq f(x_{h_t,i_t}) + \phi(h_t) + c\sqrt{\frac{2\widehat{\mathbb{V}}_{h_t,i_t}(t)\log(1/\tilde{\delta}(t^+))}{T_{h_t,i_t}(t)}} + \frac{3bc^2\log(1/\tilde{\delta}(t^+))}{T_{h_t,i_t}(t)} \\[2mm]
&\quad + c\sqrt{\frac{2\widehat{\mathbb{V}}_{h_t,i_t}(t)\log(1/\tilde{\delta}(t))}{T_{h_t,i_t}(t)}} + \frac{3bc^2\log(1/\tilde{\delta}(t))}{T_{h_t,i_t}(t)} \\[2mm]
&\leq f(x_{h_t,i_t}) + \phi(h_t) + 2c\sqrt{\frac{2\widehat{\mathbb{V}}_{h_t,i_t}(t)\log(1/\tilde{\delta}(t^+))}{T_{h_t,i_t}(t)}} + \frac{6bc^2\log(1/\tilde{\delta}(t^+))}{T_{h_t,i_t}(t)}
\end{aligned} \tag{5}$$

where the first inequality holds by the definition of $U$ and the second one holds by $t^+ \geq t$. Similarly the parent node satisfies the above inequality

$$U_{h_t^p,i_t^p}(t) \leq f(x_{h_t^p,i_t^p}) + \phi(h_t^p) + 2c\sqrt{\frac{2\widehat{\mathbb{V}}_{h_t^p,i_t^p}(t)\log(1/\tilde{\delta}(t^+))}{T_{h_t^p,i_t^p}(t)}} + \frac{6bc^2\log(1/\tilde{\delta}(t^+))}{T_{h_t^p,i_t^p}(t)}$$

By Lemma B.4, we know $U_{h^*,i^*}(t) \geq f^*$. If $(h^*,i^*)$ is a leaf, then by our definition $B_{h^*,i^*}(t) = U_{h^*,i^*}(t) \geq f^*$. Otherwise, there exists a leaf $(h_x, i_x)$ containing the maximum point which has $(h^*, i^*)$ as its ancestor. Therefore we know that $f^* \leq B_{h_x,i_x} \leq B_{h^*,i^*}$, so $B_{h^*,i^*}$ is always an upper bound for $f^*$. Now we know that

$$\Delta_{h_t,i_t}(t) := f^* - f(x_{h_t,i_t}) \leq \phi(h_t) + 2c\sqrt{\frac{2\widehat{\mathbb{V}}_{h_t,i_t}(t)\log(1/\tilde{\delta}(t^+))}{T_{h_t,i_t}(t)}} + \frac{6bc^2\log(1/\tilde{\delta}(t^+))}{T_{h_t,i_t}(t)}$$

$$\Delta_{h_t^p,i_t^p}(t) := f^* - f(x_{h_t^p,i_t^p}) \leq \phi(h_t^p) + 2c\sqrt{\frac{2\widehat{\mathbb{V}}_{h_t^p,i_t^p}(t)\log(1/\tilde{\delta}(t^+))}{T_{h_t^p,i_t^p}(t)}} + \frac{6bc^2\log(1/\tilde{\delta}(t^+))}{T_{h_t^p,i_t^p}(t)}$$

Recall that the algorithm selects a node only when $T_{h_t,i_t}(t) < \tau_{h_t,i_t}(t)$ and thus the statistical uncertainty is large, i.e., $\phi(h_t) \leq \text{SE}_{h_t,i_t}(T,t)$, and the choice of $\tau_{h_t,i_t}(t)$, we get

$$\Delta_{h_t,i_t}(t) \leq 3c\sqrt{\frac{2\widehat{\mathbb{V}}_{h_t,i_t}(t)\log(1/\tilde{\delta}(t^+))}{T_{h_t,i_t}(t)}} + \frac{9bc^2\log(1/\tilde{\delta}(t^+))}{T_{h_t,i_t}(t)}$$

$$\leq 3c\sqrt{\frac{2\widehat{\mathbb{V}}_{h_t,i_t}(t)\log(2/\tilde{\delta}(t))}{T_{h_t,i_t}(t)}} + \frac{9bc^2\log(2/\tilde{\delta}(t))}{T_{h_t,i_t}(t)}$$

where we used the fact $t^+ \leq 2t$ for any $t$. For the parent $(h_t^p, i_t^p)$, since $T_{h_t^p,i_t^p}(t) \geq \tau_{h_t^p,i_t^p}(t)$ and thus $\phi(h_t^p) \geq \text{SE}_{h_t^p,i_t^p}(T,t)$, we know that

$$\Delta_{h_t^p,i_t^p}(t) \leq \phi(h_t^p) + 2c\sqrt{\frac{2\widehat{\mathbb{V}}_{h_t^p,i_t^p}(t)\log(1/\tilde{\delta}(t^+))}{\tau_{h_t^p,i_t^p}(t)}} + \frac{6bc^2\log(1/\tilde{\delta}(t^+))}{\tau_{h_t^p,i_t^p}(t)} \leq 3\phi(h_t^p)$$

The above inequality implies that the selected node $\mathcal{P}_{h_t,i_t}$ must have a $3\phi(h_t^p)$ optimal parent under $\mathcal{E}_t$. $\square$

**Lemma B.4. ($U$ Upper Bounds $f^*$)** *Under event $\mathcal{E}_t$, we have that for any optimal node $(h^*, i^\star)$ and any choice of the smoothness function $\phi(h)$ in Algorithm 3, $U_{h^*,i^*}(t)$ is an upper bound on $f^\star$*

**Proof**. The proof here is similar to that of Lemma 5 in Shang et al. (2019). Since $t^+ \geq t$, we have

$$U_{h^*,i^*}(t) = \widehat{\mu}_{h^*,i^*}(t) + \phi(h^*) + c\sqrt{\frac{2\widehat{\mathbb{V}}_{h^*,i^*}(t)\log(1/\tilde{\delta}(t^+))}{T_{h^*,i^*}(t)}} + \frac{3bc^2\log(1/\tilde{\delta}(t^+))}{T_{h^*,i^*}(t)}$$

$$\geq \widehat{\mu}_{h^*,i^*}(t) + \phi(h^*) + c\sqrt{\frac{2\widehat{\mathbb{V}}_{h^*,i^*}(t)\log(1/\tilde{\delta}(t))}{T_{h^*,i^*}(t)}} + \frac{3bc^2\log(1/\tilde{\delta}(t))}{T_{h^*,i^*}(t)}$$

$$\geq \phi(h^*) + f(x_{h^*,i^*})$$

where the last inequality is by the event $\mathcal{E}_t$,

$$\widehat{\mu}_{h^*,i^*}(t) + c\sqrt{\frac{2\widehat{\mathbb{V}}_{h^*,i^*}(t)\log(1/\tilde{\delta}(t))}{T_{h^*,i^*}(t)}} + \frac{3bc^2\log(1/\tilde{\delta}(t))}{T_{h^*,i^*}(t)} \geq f(x_{h^*,i^*}) \qquad \square$$

**Lemma B.5. (Details for Solving $\tau$)** *For any choice of $\phi(h)$, the solution $\tau_{h,i}(t)$ to the equation $\phi(h) = \text{SE}_{h,i}(T,t)$ for the proposed VHCT algorithm in Section 4 is*

$$\tau_{h,i}(t) = \left(1 + \sqrt{1 + \frac{3b\phi(h)}{\widehat{\mathbb{V}}_{h,i}(t)/2}}\right)^2 \frac{c^2}{2\phi(h)^2}\widehat{\mathbb{V}}_{h,i}(t)\log(1/\tilde{\delta}(t^+)) \tag{6}$$

**Proof.** First, we define the following variables for ease of notatioins

$$\begin{cases} A := \phi(h) \\ B := c\sqrt{2\widehat{\mathbb{V}}_{h,i}(t)\log(1/\tilde{\delta}(t^+))} \\ C := 3bc^2\log(1/\tilde{\delta}(t^+)) \end{cases}$$

Therefore the original equation $\phi(h) = \mathrm{SE}_{h,i}(T,t)$ can be written as,

$$A = B \cdot \frac{1}{\sqrt{\tau_{h,i}(t)}} + \frac{C}{\tau_{h,i}(t)}$$

Note that the above is a quadratic equation of $\tau_{h,i}(t)$, therefore we arrive at the solution

$$\tau_{h,i}(t) = \left(1 + \sqrt{1 + \frac{3bA}{\widehat{\mathbb{V}}_{h,i}(t)/2}}\right)^2 \frac{c^2}{2A^2} \widehat{\mathbb{V}}_{h,i}(t) \log(1/\tilde{\delta}(t^+)) \qquad \qquad \square$$

## B.3 Supporting Lemmas

**Lemma B.6.** *Let $X_1, \ldots, X_t$ be i.i.d. random variables taking their values in $[\mu - \frac{b}{2}, \mu + \frac{b}{2}]$, where $\mu = \mathbb{E}[X_i]$. Let $\bar{X}_t, V_t$ be the mean and variance of $\{X_i\}_{i=1:t}$. For any $t \in \mathbb{N}$ and $x > 0$, with probability at least $1 - 3e^{-x}$, we have*

$$|\bar{X}_t - \mu| \le \sqrt{\frac{2V_t x}{t}} + \frac{3bx}{t}$$

**Proof.** This lemma follows the results in Lemma B.7. Note that $X_1, X_2, \ldots, X_t \in [\mu - \frac{b}{2}, \mu + \frac{b}{2}]$, we can define $Y_i = X_i - (\mu - \frac{b}{2})$ then $Y_1, Y_2, \ldots, Y_t \in [0, b]$ and they are *i.i.d* variables. Therefore for any $t \in \mathbb{N}$ and $x > 0$, with probability at least $1 - 3e^{-x}$, we have

$$\left|\bar{Y}_t - \frac{b}{2}\right| \le \sqrt{\frac{2V_t(Y)x}{t}} + \frac{3bx}{t}$$

Since the variance of $Y_i$ is the same as the variance of $X_i$. Therefore we have

$$|\bar{X}_t - \mu| \le \sqrt{\frac{2V_t(X)x}{t}} + \frac{3bx}{t} \qquad \qquad \square$$

**Lemma B.7. (Bernstein Inequality, Theorem 1 in Audibert et al. (2009))** *Let $X_1, \ldots, X_t$ be i.i.d. random variables taking their values in $[0, b]$. Let $\mu = \mathbb{E}[X_1]$ be their common expected value. Consider the empirical mean $\bar{X}_t$ and variance $V_t$ defined respectively by*

$$\bar{X}_t = \frac{\sum_{i=1}^{t} X_i}{t} \quad and \quad V_t = \frac{\sum_{i=1}^{t} \left(X_i - \bar{X}_t\right)^2}{t}$$

*Then, for any $t \in \mathbb{N}$ and $x > 0$, with probability at least $1 - 3e^{-x}$*

$$|\bar{X}_t - \mu| \le \sqrt{\frac{2V_t x}{t}} + \frac{3bx}{t}$$

*Furthermore, introducing*

$$\beta(x, t) = 3 \inf_{1 < \alpha \le 3} \left(\frac{\log t}{\log \alpha} \wedge t\right) e^{-x/\alpha}$$

*where $u \wedge v$ denotes the minimum of $u$ and $v$, we have for any $t \in \mathbb{N}$ and $x > 0$ with probability at least $1 - \beta(x, t)$*

$$|\bar{X}_s - \mu| \le \sqrt{\frac{2V_s x}{s}} + \frac{3bx}{s}$$

*holds simultaneously for $s \in \{1, 2, \ldots, t\}$.*

## C  Proof of Theorem 4.1

### C.1  The choice of $\tau_{h,i}(t)$.

When $\phi(h) = \nu_1 \rho^h$, we have the following choice of $\tau_{h,i}(t)$ by Lemma B.5.

$$
\begin{aligned}
\tau_{h,i}(t) &= \left(1 + \sqrt{1 + \frac{3b\nu_1\rho^h}{\widehat{\mathbb{V}}_{h,i}(t)/2}}\right)^2 \left(\frac{c}{\nu_1\rho^h}\right)^2 (\widehat{\mathbb{V}}_{h,i}(t)/2) \cdot \log(1/\tilde{\delta}(t^+)) \\
&= \left(2 + 2\sqrt{1 + \frac{3b\nu_1\rho^h}{\widehat{\mathbb{V}}_{h,i}(t)/2}} + \frac{3b\nu_1\rho^h}{\widehat{\mathbb{V}}_{h,i}(t)/2}\right) \frac{c^2 \log\left(1/\tilde{\delta}\left(t^+\right)\right)(\widehat{\mathbb{V}}_{h,i}(t)/2)}{\nu_1^2} \rho^{-2h} \\
&= \left(\widehat{\mathbb{V}}_{h,i}(t) + \sqrt{\widehat{\mathbb{V}}_{h,i}(t)^2 + 6b\nu_1\rho^h\widehat{\mathbb{V}}_{h,i}(t)} + 3b\nu_1\rho^h\right) \frac{c^2 \log\left(1/\tilde{\delta}\left(t^+\right)\right)}{\nu_1^2} \rho^{-2h}
\end{aligned}
\tag{7}
$$

Since variance is non-negative, we have $\tau_{h,i}(t) \geq \frac{3bc^2}{\nu_1}\rho^{-h}$. When the variance term $\widehat{\mathbb{V}}_{h,i}(t)$ is small, the other two terms are small. We also have the following upper bound for $\tau_{h,i}(t)$.

$$
\tau_{h,i}(t) \leq D_1^2 \frac{c^2 \log\left(1/\tilde{\delta}\left(t^+\right)\right)}{\nu_1^2} \rho^{-2h} + 3b\nu_1 \frac{c^2 \log\left(1/\tilde{\delta}\left(t^+\right)\right)}{\nu_1^2} \rho^{-h}
$$

where we define the constant $D_1^2 = \left(V_{\max} + 2\sqrt{V_{\max}^2 + 6bV_{\max}\nu_1}\right) = \mathcal{O}(V_{\max})$.

### C.2  Main proof

This part of the proof follows Theorem 3.1. Let $\overline{H}$ be an integer that satisfies $1 \leq \overline{H} < H(n)$ to be decided later.

$$
\begin{aligned}
\widetilde{R}_n^{\mathcal{E}} = \sum_{t=1}^{n} \Delta_{h_t,i_t} \mathbb{I}_{\mathcal{E}_t} &\leq \sum_{h=0}^{H(n)} \sum_{i \in \mathcal{I}_h(n)} \sum_{t=1}^{n} \Delta_{h,i} \mathbb{I}_{(h_t,i_t)=(h,i)} \mathbb{I}_{\mathcal{E}_t} \\
&\leq \underbrace{2aC \sum_{h=1}^{\overline{H}} (\mathtt{OE}_{h-1})^{-\bar{d}} \sum_{t=1}^{n} \max_{i \in \mathcal{I}_h(n)} \mathtt{SE}_{h,i}(T,t)}_{(a)} + \underbrace{a \sum_{\overline{H}+1}^{H(n)} \sum_{i \in \mathcal{I}_h(n)} \sum_{t=1}^{n} \mathtt{SE}_{h,i}(T,t)}_{(b)}
\end{aligned}
$$

By Lemma B.3, we have $a = 3$ and thus the following inequality

$$
\begin{aligned}
\widetilde{R}_n^{\mathcal{E}} &\leq \underbrace{\sum_{h=0}^{\overline{H}} 2C\rho^{-d(h-1)} \left\{6c\sqrt{2(\max_{i \in \mathcal{I}_h(n)}\{\tau_{h,i}(n)\})V_{\max}\log(2/\tilde{\delta}(n))} + 9bc^2\log(2/\tilde{\delta}(n))\log(\max_{i \in \mathcal{I}_h(n)}\{\tau_{h,i}(n)\})\right\}}_{(a)} \\
&+ \underbrace{\sum_{h=\overline{H}+1}^{H(n)} \sum_{i \in \mathcal{I}_h(n)} \left\{6c\sqrt{2T_{h,i}(n)V_{\max}\log(2/\tilde{\delta}(\bar{t}_{h,i}))} + 9bc^2\log(2/\tilde{\delta}(\bar{t}_{h,i}))\log T_{h,i}(n)\right\}}_{(b)}
\end{aligned}
$$

Now we bound the two terms $(a)$ and $(b)$ of $\widetilde{R}_n^{\mathcal{E}}$ separately.

$$(a) \leq \sum_{h=0}^{\overline{H}} 2C\rho^{-d(h-1)} \left\{ 6c\sqrt{2(\max_{i\in\mathcal{I}_h(n)}\{\tau_{h,i}(n)\})V_{\max}\log(2/\tilde{\delta}(n))} + 9bc^2\log(2/\tilde{\delta}(n))\log(\max_{i\in\mathcal{I}_h(n)}\{\tau_{h,i}(n)\}) \right\}$$

$$\leq \sum_{h=0}^{\overline{H}} \frac{2CD_1c\rho^d\log(2/\tilde{\delta}(n))}{\nu_1}6c\sqrt{2V_{\max}}\rho^{-h(d+1)} + \frac{2C\sqrt{3b\nu_1}c\rho^d\log(2/\tilde{\delta}(n))}{\nu_1}6c\sqrt{2V_{\max}}\rho^{-h(d+\frac{1}{2})}$$

$$+ 18C\rho^{-d(h-1)}bc^2\log(2/\tilde{\delta}(n))\left(\log\left(\log(2/\tilde{\delta}(n))\right) + 2\log(\frac{D_1c}{\nu_1}) - 2h\log\rho\right)$$

$$\leq \frac{12\sqrt{2V_{\max}}CD_1c^2\rho^d\log(2/\tilde{\delta}(n))}{\nu_1(1-\rho)}\rho^{-\overline{H}(d+1)} + \frac{12\sqrt{6b\nu_1 V_{\max}}Cc^2\rho^d\log(2/\tilde{\delta}(n))}{\nu_1(1-\rho)}\rho^{-\overline{H}(d+\frac{1}{2})}$$

$$+ \frac{18Cbc^2\rho^{2d}\log(2/\tilde{\delta}(n))\left(\log\left(\log(2/\tilde{\delta}(n))\right) + 2\log(\frac{D_1c}{\nu_1})\right)}{1-\rho}\rho^{-\overline{H}d}$$

$$+ 36Cbc^2\log(2/\tilde{\delta}(n))\log(\frac{1}{\rho})\frac{1}{(1-\rho^d)^2}\left((\rho^d\overline{H} - \rho^{2d}\overline{H} - \rho^{2d})\rho^{-d\overline{H}} + \rho^{2d}\right)$$

where in the second inequality we used the upper bound of $\tau_h(t)$ in Section B. The last inequality is by the formula for the sum of a geometric sequence and the following result.

$$\sum_{h=0}^{\overline{H}} h\rho^{-d(h-1)} = \frac{1}{1-\rho^d}\left(\overline{H}\rho^{-d(\overline{H}-1)} - \sum_{h=-1}^{\overline{H}-2}\rho^{-dh}\right)$$

$$= \frac{1}{(1-\rho^d)^2}\left((\rho^d\overline{H} - \rho^{2d}\overline{H} - \rho^{2d})\rho^{-d\overline{H}} + \rho^{2d}\right)$$

$$\leq \frac{1}{(1-\rho)^2}\left((\rho^d\overline{H} - \rho^{2d}\overline{H} - \rho^{2d})\rho^{-d\overline{H}} + \rho^{2d}\right)$$

Next we bound the second term $(b)$ in the summation. By the Cauchy-Schwarz Inequality,

$$(b) \leq \sum_{h=\overline{H}+1}^{H(n)}\sum_{i\in\mathcal{I}_h(n)} \left\{ 6c\sqrt{2T_{h,i}(n)V_{\max}\log(2/\tilde{\delta}(\bar{t}_{h,i}))} + 9bc^2\log(2/\tilde{\delta}(\bar{t}_{h,i}))\log T_{h,i}(n) \right\}$$

$$\leq \sqrt{n\sum_{h=\overline{H}+1}^{H(n)}\sum_{i\in\mathcal{I}_h(n)}\log(2/\tilde{\delta}(\bar{t}_{h,i}))} + \sum_{h=\overline{H}+1}^{H(n)}\sum_{i\in\mathcal{I}_h(n)}9bc^2\log(2/\tilde{\delta}(\bar{t}_{h,i}))\log T_{h,i}(n)$$

Recall that our algorithm only selects a node when $T_{h,i}(t) \geq \tau_{h,i}(t)$ for its parent, i.e. when the number of pulls is larger than the threshold and the algorithm finds the node by passing its parent. Therefore we have

$$T_{h,i}(\tilde{t}_{h,i}) \geq \tau_{h,i}(\tilde{t}_{h,i}), \forall h \in [0, H(n) - 1], i \in \mathcal{I}_h(n)^+$$

So we have the following set of inequalities.

$$n = \sum_{h=0}^{H(n)} \sum_{i \in \mathcal{I}_h(n)} T_{h,i}(n) \geq \sum_{h=0}^{H(n)-1} \sum_{i \in \mathcal{I}_h^+(n)} T_{h,i}(n) \geq \sum_{h=0}^{H(n)-1} \sum_{i \in \mathcal{I}_h^+(n)} T_{h,i}(\widetilde{t}_{h,i}) \geq \sum_{h=0}^{H(n)-1} \sum_{i \in \mathcal{I}_h^+(n)} \tau_{h,i}(\widetilde{t}_{h,i})$$

$$\geq \sum_{h=\overline{H}}^{H(n)-1} \sum_{i \in \mathcal{I}_h^+(n)} \frac{c^2 \log\left(1/\widetilde{\delta}\left(t^+\right)\right) \epsilon}{\nu_1^2} \rho^{-2h} \geq c^2 \rho^{-2\overline{H}} \epsilon \sum_{h=\overline{H}}^{H(n)-1} \sum_{i \in \mathcal{I}_h^+(n)} \frac{\log\left(1/\widetilde{\delta}\left(\widetilde{t}_{h,i}^+\right)\right)}{\nu_1^2}$$

$$= c^2 \rho^{-2\overline{H}} \epsilon \sum_{h=\overline{H}}^{H(n)-1} \sum_{i \in \mathcal{I}_h^+(n)} \frac{\log\left(1/\widetilde{\delta}\left(\max[\overline{t}_{h+1,2i-1}, \overline{t}_{h+1,2i}]^+\right)\right)}{\nu_1^2}$$

$$= c^2 \rho^{-2\overline{H}} \epsilon \sum_{h=\overline{H}}^{H(n)-1} \sum_{i \in \mathcal{I}_h^+(n)} \frac{\max[\log\left(1/\widetilde{\delta}\left(\overline{t}_{h+1,2i-1}^+\right)\right), \log\left(1/\widetilde{\delta}\left(\overline{t}_{h+1,2i}^+\right)\right)]}{\nu_1^2}$$

$$\geq c^2 \rho^{-2\overline{H}} \epsilon \sum_{h=\overline{H}}^{H(n)-1} \sum_{i \in \mathcal{I}_h^+(n)} \frac{\log\left(1/\widetilde{\delta}\left(\overline{t}_{h+1,2i-1}^+\right)\right) + \log\left(1/\widetilde{\delta}\left(\overline{t}_{h+1,2i}^+\right)\right)}{2\nu_1^2}$$

$$= c^2 \rho^{-2\overline{H}} \epsilon \sum_{h=\overline{H}+1}^{H(n)} \sum_{i \in \mathcal{I}_{h-1}^+(n)} \frac{\log\left(1/\widetilde{\delta}\left(\overline{t}_{h,2i-1}^+\right)\right) + \log\left(1/\widetilde{\delta}\left(\overline{t}_{h,2i}^+\right)\right)}{2\nu_1^2}$$

$$= \frac{c^2 \rho^{-2\overline{H}}}{2\nu_1^2} \epsilon \sum_{h=\overline{H}+1}^{H(n)} \sum_{i \in \mathcal{I}_h(n)} \log\left(1/\widetilde{\delta}\left(\overline{t}_{h,i}^+\right)\right)$$

Note that in the second equality, we have used the definition of $\widetilde{t}_{h,i}$, $\widetilde{t}_{h,i} = \max(\overline{t}_{h,i}, \overline{t}_{h,i})$. Moreover, the third equality relies on the following fact

$$\log\left(1/\widetilde{\delta}\left(\max\left\{\overline{t}_{h+1,2i-1}, \overline{t}_{h+1,2i}\right\}^+\right)\right) = \max\left\{\log\left(1/\widetilde{\delta}\left(\overline{t}_{h+1,2i-1}^+\right)\right), \log\left(1/\widetilde{\delta}\left(\overline{t}_{h+1,2i}^+\right)\right)\right\}$$

The next equality is just by change of variables $h = h + 1$. In the last inequality, we used the fact that for any $h > 0$, $\mathcal{I}_h^+(n)$ covers all the internal nodes at level $h$, so the set of the children of $I_h^+(n)$ covers $I_{h+1}(n)$. In other words, we have proved that

$$\sum_{h=\overline{H}+1}^{H(n)} \sum_{i \in \mathcal{I}_h(n)} \log\left(1/\widetilde{\delta}\left(\overline{t}_{h,i}^+\right)\right) \leq \frac{2n\nu_1^2 \rho^{2\overline{H}}}{c^2 \epsilon}$$

Therefore we have

$$(b) \leq 2n \frac{\nu_1 \rho^{\overline{H}}}{c\sqrt{\epsilon}} + \frac{18b\nu_1^2 \rho^{2\overline{H}} n \log n}{\epsilon}$$

If we let the dominating terms in (a) and (b) be equal, then

$$\rho^{\overline{H}} = \left(\frac{12\sqrt{2V_{\max}} C D_1 c^3 \sqrt{\epsilon} \rho^d \log(2/\widetilde{\delta}(n))}{2\nu_1^2 n(1-\rho)}\right)^{\frac{1}{d+2}}$$

Substitute the above choice of $\rho^{\overline{H}}$ into the original inequality, then the dominating terms in (a) and (b) reduce to $\widetilde{\mathcal{O}}(C_1 V_{\max}^{\frac{1}{d+2}} n^{\frac{d+1}{d+2}})$ because $D_1 = \Theta(\sqrt{V_{\max}})$, where $C_1$ is a constant that does not depend on the variance. The non-dominating terms are all $\widetilde{\mathcal{O}}(n^{\frac{2d+1}{2d+4}})$, we get

$$\widetilde{R}_n^{\mathcal{E}} \leq (a) + (b) \leq C_1 V_{\max}^{\frac{1}{d+2}} n^{\frac{d+1}{d+2}} (\log \frac{n}{\delta})^{\frac{1}{d+2}} + C_2 n^{\frac{2d+1}{2d+4}} \log \frac{n}{\delta} \tag{8}$$

where $C_2$ is another constant. Finally, combining all the results in Theorem 3.1, Lemma B.2, Eqn. (8), we can obtain the upper bound

$$\widetilde{R}_n^{\texttt{VHCT}} \leq \sqrt{n} + \sqrt{2n \log(\frac{4n^2}{\delta})} + C_1 V_{\max}^{\frac{1}{d+2}} n^{\frac{d+1}{d+2}} (\log \frac{n}{\delta})^{\frac{1}{d+2}} + C_2 n^{\frac{2d+1}{2d+4}} \log \frac{n}{\delta}$$

$$\leq 2\sqrt{2n \log(\frac{4n^2}{\delta})} + C_1 V_{\max}^{\frac{1}{d+2}} n^{\frac{d+1}{d+2}} (\log \frac{n}{\delta})^{\frac{1}{d+2}} + C_2 n^{\frac{2d+1}{2d+4}} \log \frac{n}{\delta}$$

The expectation in the theorem can be shown by directly taking $\delta = 1/n$ as in Theorem 3.1. $\qquad\square$

## D    Proof of Theorem 4.2

### D.1    Choice of the Threshold

By Lemma B.5, we get that when $\phi(h) = 1/h$, we can solve for $\tau_h$ as follows.

$$\tau_{h,i}(t) = \left(1 + \sqrt{1 + \frac{3b/h}{\widehat{\mathbb{V}}_{h,i}(t)(t)/2}}\right)^2 c^2 h^2 (\widehat{\mathbb{V}}_{h,i}(t)(t)/2) \cdot \log(1/\tilde{\delta}(t^+))$$

$$= \left(\widehat{\mathbb{V}}_{h,i}(t) + \sqrt{\widehat{\mathbb{V}}_{h,i}(t)^2 + 6b/h\widehat{\mathbb{V}}_{h,i}(t)} + 3b/h\right) c^2 \log\left(1/\tilde{\delta}\left(t^+\right)\right) h^2$$

$$\leq D_1^2 c^2 \log\left(1/\tilde{\delta}\left(t^+\right)\right) h^2 + 3bc^2 \log\left(1/\tilde{\delta}\left(t^+\right)\right) h$$

where we again define a new constant $D_1^2 = \left(V_{\max} + 2\sqrt{V_{\max}^2 + 6bV_{\max}}\right) = \Theta(V_{\max})$.

### D.2    Main Proof

The failing confidence interval part can be easily done as in Section C since $\mathcal{E}_t$ is also a high-probability event at each time $t$. We start from the bound on $\widetilde{R}^{\mathcal{E}}$. By Theorem 3.1 and similar to what we have done in Theorem 4.1, we decompose $\widetilde{R}^{\mathcal{E}}$ over different depths. Let $1 \leq \overline{H} < H(n)$ be an integer to be decided later, then we have

$$\widetilde{R}_n^{\mathcal{E}} = \sum_{t=1}^{n} \Delta_{h_t,i_t} \mathbb{I}_{\mathcal{E}_t} \leq \sum_{h=0}^{H(n)} \sum_{i \in \mathcal{I}_h(n)} \sum_{t=1}^{n} \Delta_{h,i} \mathbb{I}_{(h_t,i_t)=(h,i)} \mathbb{I}_{\mathcal{E}_t}$$

$$\leq \underbrace{2aC \sum_{h=1}^{\overline{H}} (\texttt{OE}_{h-1})^{-\bar{d}} \sum_{t=1}^{n} \max_{i \in \mathcal{I}_h(n)} \texttt{SE}_{h,i}(T,t)}_{(a)} + \underbrace{a \sum_{\overline{H}+1}^{H(n)} \sum_{i \in \mathcal{I}_h(n)} \sum_{t=1}^{n} \texttt{SE}_{h,i}(T,t)}_{(b)}$$

Now we bound the two terms $(a)$ and $(b)$ of $\widetilde{R}_n^{\mathcal{E}}$ separately. By Lemma B.3, we have $a = 3$.

$$(a) \leq \sum_{h=0}^{\overline{H}} 2Ch^{\bar{d}} \left\{ 6c\sqrt{2\max_i\{\tau_{h,i}(n)\}V_{\max}\log(2/\tilde{\delta}(n))} + 9bc^2\log(2/\tilde{\delta}(n))\log(\max_{i \in \mathcal{I}_h(n)}\{\tau_{h,i}(n)\}) \right\}$$

$$\leq \sum_{h=0}^{\overline{H}} 12CD_1c^2h^{\bar{d}+1}\sqrt{2\log\left(1/\tilde{\delta}(n)\right)V_{\max}\log(2/\tilde{\delta}(n))}$$

$$+ \sum_{h=0}^{\overline{H}} 36bCc^2h^{\bar{d}+\frac{1}{2}}\sqrt{2\log\left(1/\tilde{\delta}(n)\right)V_{\max}\log(2/\tilde{\delta}(n))}$$

$$+ 18Cbc^2h^{\bar{d}}\log(2/\tilde{\delta}(n))\log\left(D_1^2c^2\log\left(2/\tilde{\delta}(n)\right)h^2\right)$$

$$\leq \sum_{h=0}^{\overline{H}} 12aCD_1c^2h^{\bar{d}+1}\log(2/\tilde{\delta}(n))\sqrt{2V_{\max}} + \sum_{h=0}^{\overline{H}} 36bCc^2h^{\bar{d}+\frac{1}{2}}\log(2/\tilde{\delta}(n))\sqrt{2V_{\max}}$$

$$+ \sum_{h=0}^{\overline{H}} 18Cbc^2h^{\bar{d}}\log(2/\tilde{\delta}(n))\log\left(D_1^2c^2\log\left(2/\tilde{\delta}(n)\right)n^2\right)$$

$$\leq 12\sqrt{2V_{\max}}aCD_1c^2\log(2/\tilde{\delta}(n))\sum_{h=0}^{\overline{H}} h^{\bar{d}+1} + 36bCc^2\log(2/\tilde{\delta}(n))\sqrt{2V_{\max}}\sum_{h=0}^{\overline{H}} h^{\bar{d}+\frac{1}{2}}$$

$$+ 18Cbc^2\log(2/\tilde{\delta}(n))\log\left(D_1^2c^2\log\left(2/\tilde{\delta}(n)\right)n^2\right)\sum_{h=0}^{\overline{H}} h^{\bar{d}}$$

$$\leq 12\sqrt{2V_{\max}}aCD_1c^2\log(2/\tilde{\delta}(n))\left(\sum_{h=0}^{\overline{H}} h\right)^{\bar{d}+1} + 36bCc^2\log(2/\tilde{\delta}(n))\sqrt{2V_{\max}}\left(\sum_{h=0}^{\overline{H}} h\right)^{\bar{d}+\frac{1}{2}}$$

$$+ 18Cbc^2\log(2/\tilde{\delta}(n))\log\left(D_1^2c^2\log\left(2/\tilde{\delta}(n)\right)n^2\right)\left(\sum_{h=0}^{\overline{H}} h\right)^{\bar{d}}$$

$$\leq 12\sqrt{2V_{\max}}aCD_1c^2\log(2/\tilde{\delta}(n))\left(\frac{\overline{H}(\overline{H}+1)}{2}\right)^{\bar{d}+1} + 36bCc^2\log(2/\tilde{\delta}(n))\sqrt{2V_{\max}}\left(\frac{\overline{H}(\overline{H}+1)}{2}\right)^{\bar{d}+\frac{1}{2}}$$

$$+ 18Cbc^2\log(2/\tilde{\delta}(n))\log\left(D_1^2c^2\log\left(2/\tilde{\delta}(n)\right)n^2\right)\left(\frac{\overline{H}(\overline{H}+1)}{2}\right)^{\bar{d}}$$

Next we bound the second term $(b)$ in the summation. By the Cauchy-Schwarz Inequality,

$$(b) \leq \sum_{h=\overline{H}+1}^{H(n)} \sum_{i \in \mathcal{I}_h(n)} \left\{ 6c\sqrt{2T_{h,i}(n)V_{\max}\log(2/\tilde{\delta}(\bar{t}_{h,i}))} + 9bc^2\log(2/\tilde{\delta}(\bar{t}_{h,i}))\log T_{h,i}(n) \right\}$$

$$\leq \sqrt{n\sum_{h=\overline{H}+1}^{H(n)}\sum_{i \in \mathcal{I}_h(n)}\log(2/\tilde{\delta}(\bar{t}_{h,i}))} + \sum_{h=\overline{H}+1}^{H(n)}\sum_{i \in \mathcal{I}_h(n)} 9bc^2\log(2/\tilde{\delta}(\bar{t}_{h,i}))\log T_{h,i}(n)$$

Recall that our algorithm only selects a node when $T_{h,i}(t) \geq \tau_{h,i}(t)$ for its parent, i.e. when the number of pulls is larger than the threshold and the algorithm finds the node by passing its parent. Therefore we have

$$T_{h,i}(\widetilde{t}_{h,i}) \geq \tau_{h,i}(\widetilde{t}_{h,i}), \forall h \in [0, H(n)-1], i \in \mathcal{I}_h(n)^+$$

So we have the following set of inequalities.

$$n = \sum_{h=0}^{H(n)} \sum_{i \in \mathcal{I}_h(n)} T_{h,i}(n) \geq \sum_{h=0}^{H(n)-1} \sum_{i \in \mathcal{I}_h^+(n)} T_{h,i}(n) \geq \sum_{h=0}^{H(n)-1} \sum_{i \in \mathcal{I}_h^+(n)} T_{h,i}(\widetilde{t}_{h,i}) \geq \sum_{h=0}^{H(n)-1} \sum_{i \in \mathcal{I}_h^+(n)} \tau_{h,i}(\widetilde{t}_{h,i})$$

$$\geq \sum_{h=\overline{H}}^{H(n)-1} \sum_{i \in \mathcal{I}_h^+(n)} c^2 \log\left(1/\widetilde{\delta}\left(t^+\right)\right) \epsilon h^2 \geq c^2 \overline{H}^2 \epsilon \sum_{h=\overline{H}}^{H(n)-1} \sum_{i \in \mathcal{I}_h^+(n)} \log\left(1/\widetilde{\delta}\left(\overline{t}_{h,i}^+\right)\right)$$

$$= c^2 \overline{H}^2 \epsilon \sum_{h=\overline{H}}^{H(n)-1} \sum_{i \in \mathcal{I}_h^+(n)} \log\left(1/\widetilde{\delta}\left(\max[\overline{t}_{h+1,2i-1}, \overline{t}_{h+1,2i}]^+\right)\right)$$

$$= c^2 \overline{H}^2 \epsilon \sum_{h=\overline{H}}^{H(n)-1} \sum_{i \in \mathcal{I}_h^+(n)} \max[\log\left(1/\widetilde{\delta}\left(\overline{t}_{h+1,2i-1}^+\right)\right), \log\left(1/\widetilde{\delta}\left(\overline{t}_{h+1,2i}^+\right)\right)]$$

$$\geq \frac{c^2 \overline{H}^2 \epsilon}{2} \sum_{h=\overline{H}}^{H(n)-1} \sum_{i \in \mathcal{I}_h^+(n)} \log\left(1/\widetilde{\delta}\left(\overline{t}_{h+1,2i-1}^+\right)\right) + \log\left(1/\widetilde{\delta}\left(\overline{t}_{h+1,2i}^+\right)\right)$$

$$= \frac{c^2 \overline{H}^2 \epsilon}{2} \sum_{h=\overline{H}+1}^{H(n)} \sum_{i \in \mathcal{I}_{h-1}^+(n)} \log\left(1/\widetilde{\delta}\left(\overline{t}_{h,2i-1}^+\right)\right) + \log\left(1/\widetilde{\delta}\left(\overline{t}_{h,2i}^+\right)\right)$$

$$= \frac{c^2 \overline{H}^2 \epsilon}{2} \epsilon \sum_{h=\overline{H}+1}^{H(n)} \sum_{i \in \mathcal{I}_h(n)} \log\left(1/\widetilde{\delta}\left(\overline{t}_{h,i}^+\right)\right)$$

Note that in the second equality, we have used the definition of $\widetilde{t}_{h,i} = \max(\overline{t}_{h,i}, \overline{t}_{h,i})$. Moreover, the third equality relies on the following fact

$$\log\left(1/\widetilde{\delta}\left(\max\left\{\overline{t}_{h+1,2i-1}, \overline{t}_{h+1,2i}\right\}^+\right)\right) = \max\left\{\log\left(1/\widetilde{\delta}\left(\overline{t}_{h+1,2i-1}^+\right)\right), \log\left(1/\widetilde{\delta}\left(\overline{t}_{h+1,2i}^+\right)\right)\right\}$$

The next equality is just by change of variables $h = h + 1$. In the last inequality, we used the fact that for any $h > 0$, $\mathcal{I}_h^+(n)$ covers all the internal nodes at level $h$, so the set of the children of $I_h^+(n)$ covers $I_{h+1}(n)$. In other words, we have proved that

$$\sum_{h=\overline{H}+1}^{H(n)} \sum_{i \in \mathcal{I}_h(n)} \log\left(1/\widetilde{\delta}\left(\overline{t}_{h,i}^+\right)\right) \leq \frac{2n}{c^2 \epsilon} \overline{H}^{-2}$$

Therefore we have the following inequality

$$(b) \leq \frac{2n}{c\sqrt{\epsilon}} \overline{H}^{-1} + \frac{18bn \log n}{\epsilon} \overline{H}^{-2}$$

If we let the dominating terms in (a) and (b) be equal, then

$$\overline{H}^{-1} = \left(\frac{12\sqrt{2V_{\max}} C D_1 c^3 \sqrt{\epsilon} \log(2/\widetilde{\delta}(n))}{2n}\right)^{\frac{1}{2\overline{d}+3}}$$

Substitute the above choice of $\overline{H}$ into the original inequality, then the dominating terms in (a) and (b) reduce to $\widetilde{\mathcal{O}}(C_1 V_{\max}^{\frac{1}{2\overline{d}+3}} n^{\frac{2\overline{d}+2}{2\overline{d}+3}})$ because $D_1 = \Theta(\sqrt{V_{\max}})$, where $C_1$ is a constant that does not depend on the variance. The non-dominating terms are all $\widetilde{\mathcal{O}}(n^{\frac{2\overline{d}+1}{2\overline{d}+3}})$, we get

$$\widetilde{R}_n^{\mathcal{E}} \leq (a) + (b) \leq C_1 V_{\max}^{\frac{1}{2\bar{d}+3}} n^{\frac{2\bar{d}+2}{2\bar{d}+3}} (\log \frac{n}{\delta})^{\frac{1}{2\bar{d}+3}} + C_2 n^{\frac{2\bar{d}+1}{2\bar{d}+3}} \log \frac{n}{\delta} \tag{9}$$

where $C_2$ is another constant. Finally, combining all the results in Theorem 3.1, Lemma B.2, Eqn. (9), we can obtain the upper bound

$$\widetilde{R}_n^{\text{VHCT}} \leq \sqrt{n} + \sqrt{2n \log(\frac{4n^2}{\delta})} + C_1 V_{\max}^{\frac{1}{4\bar{d}+6}} n^{\frac{2\bar{d}+2}{2\bar{d}+3}} (\log \frac{n}{\delta})^{\frac{1}{2\bar{d}+3}} + C_2 n^{\frac{2\bar{d}+1}{2\bar{d}+3}} \log \frac{n}{\delta}$$

$$\leq 2\sqrt{2n \log(\frac{4n^2}{\delta})} + C_1 V_{\max}^{\frac{1}{4\bar{d}+6}} n^{\frac{2\bar{d}+2}{2\bar{d}+3}} (\log \frac{n}{\delta})^{\frac{1}{2\bar{d}+3}} + C_2 n^{\frac{2\bar{d}+1}{2\bar{d}+3}} \log \frac{n}{\delta}$$

The expectation in the theorem can be shown by directly taking $\delta = 1/n$ as in Theorem 3.1. □

## E    Experiment Details

In this appendix, we provide more experiment details and additional experiments as a supplement to Section 5. For the implementation of all the algorithms, we utilize the publicly available code of `POO` and `HOO` at the link `https://rdrr.io/cran/OOR/man/POO.html` and the PyXAB library (Li et al., 2023). For all the experiments in Section 5 and Appendix E.2, we have used a low-noise setting where $\epsilon_t \sim \text{Uniform}(-0.05, 0.05)$ to verify the advantage of `VHCT`.

### E.1    Experimental Settings

**Remarks on Bayesian Optimization Algorithm.** For the implementation of the Bayesian Optimization algorithm `BO`, we have used the publicly available code at `https://github.com/SheffieldML/GPyOpt`, which is also recommended by Frazier (2018). For the acquisition function and the prior of `BO`, we have used the default choices in the aforementioned package. We emphasize that `BO` is much more **computationally expensive** compared with the other algorithms due to the high computational complexity of Gaussian Process. The other algorithms in this paper (`HOO, HCT, VHCT`, etc.) take at most minutes to reach the endpoint of every experiment, where as `BO` typically needs a few days to finish. Moreover, the performance (cumulative regret) of `BO` is not comparable with our algorithm.

**Synthetic Experiments.** In Figure 4, we provide the performances of the different algorithms (`VHCT, HCT, T-HOO`) that need the smoothness parameters under different parameter settings $\rho \in \{0.25, 0.5, 0.75\}$. Here, we choose to plot an equivalent notion, the average regret $R_t/t$ instead of the cumulative regret $R_t$ because some curves have very large cumulative regrets, so it would be hard to compare them with the other curves. In general, $\rho = 0.75$ or $\rho = 0.5$ are good choices for `VHCT` and `HCT`, and $\rho = 0.25$ is a good choice for `T-HOO`. Therefore, we use these parameter settings in the real-life experiments and the additional experiments in the next subsection. For `POO` and `PCT`, we follow Grill et al. (2015) and use $\rho_{\max} = 0.9$. The unknown bound $b$ is set to be $b = 1$ for all the algorithms used in the experiments.

**Landmine Dataset**. The landmine dataset contains 29 landmine fields, with each field consisting of different number of positions. Each position has some features extracted from radar images, and machine learning models (like SVM) are used to learn the features and detect whether a certain position has landmine or not. The dataset is available at `http://www.ee.duke.edu/~lcarin/LandmineData.zip`. We have followed the open-source implementation at `https://github.com/daizhongxiang/Federated_Bayesian_Optimization` to process the data and train the SVM model. We tune two hyper-parameters when training SVM, the RBF kernel parameter from [0.01, 10], and the $L_2$ regularization from [1e-4, 10]. The model is trained on the training set with the selected hyper-parameter and then evaluated on the testing set. The testing AUC-ROC score is the blackbox objective to be optimized.

**MNIST Dataset and Neural Network.** The MNIST dataset can be downloaded from `http://yann.lecun.com/exdb/mnist/` and is one of the most famous image classification datasets. We have used stochastic gradient descent (SGD) to train a two-layer feed-forward neural network on the training images, and the

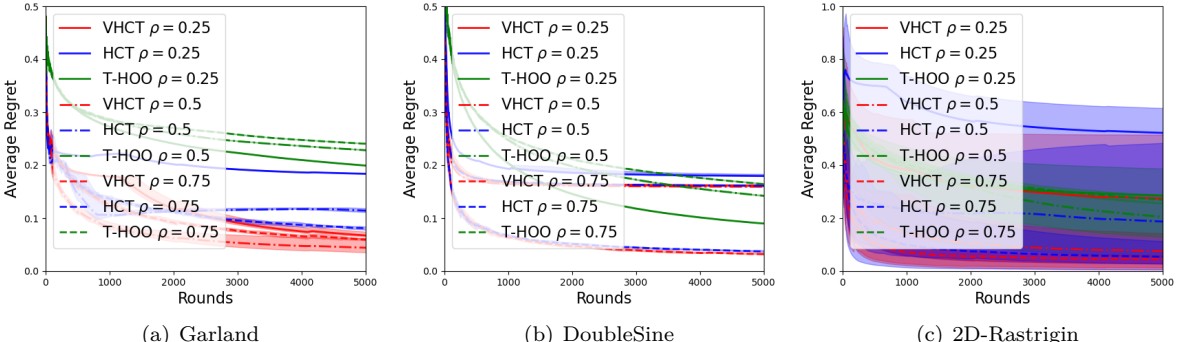

(a) Garland  (b) DoubleSine  (c) 2D-Rastrigin

Figure 4: Best parameters on the Garland, DoubleSine and Rastrigin function

objective is the validation accuracy on the testing images. We have used ReLU activation and the hidden layer has 64 units. We tune three different hyper-parameters of SGD to find the best hyper-parameter, specifically, the mini batch-size from [1, 100], the learning rate from [1e-6, 1], and the weight decay from [1e-6, 5e-1].

## E.2 Additional Experiments

In this subsection, we provide additional experiments on some other nonconvex optimization evaluation benchmarks. They are used in many optimization researches to evaluate the performance of different optimization algorithms, including the convergence rate, the precision and the robustness such as Azar et al. (2014); Shang et al. (2019); Bartlett et al. (2019). Detailed discussions of these functions can be found at `https://en.wikipedia.org/wiki/Test_functions_for_optimization` Although some of these function values (e.g., the Himmelblau function) are not bounded in $[0, 1]$ on the domain we select, the convergence rate of different algorithms will not change as long as the function is uniformly bounded over its domain. To commit a fair comparison (i.e., sharing similar signal/noise ratio), **we have re-scaled all the objectives listed below to be bounded by** $[-1, 1]$.

We list the functions used and their mathematical expressions as follows.

- DoubleSine (Figure 5(a)) is (originally) a one dimensional function proposed by Grill et al. (2015) with multiple sharp local minimums and one global minimum. The results are shown in Figure 1(a).

$$f(x, y) = 20 \exp\left[-0.2\sqrt{0.5\left(x^2 + y^2\right)}\right] + \exp[0.5(\cos 2\pi x + \cos 2\pi y)] - e - 20.$$

- The counter example $f(x) = 1 + 1/\ln x$ in 4 decreases too fast around zero and thus its smoothness cannot be measured by $\nu_1 \rho^h$ for any constants $\nu_1, \rho > 0$. However, because it is continuously differentiable and even monotone, the function is very easy to optimize. The results are shown in Figure 6(b).

- Himmelbalu (Figure 5(c)) is (originally) a two dimensional function with four flat global minimums. We use the negative of the original function for maximization, and we restrain $x$ to be in $[-5, 5]^2$ to include all four global maximums. The results are shown in Figure 6(c).

$$f(x, y) = -\left(x^2 + y - 11\right)^2 - \left(x + y^2 - 7\right)^2.$$

- Rastrigin (Figure 5(d)) is a multi-dimensional function with a vast number of sharp local minimums and one global minimum. We use the negative of the original function for maximization. We run

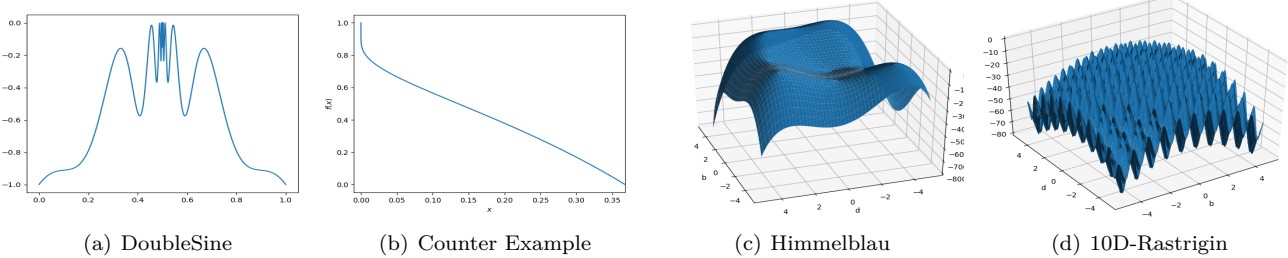

(a) DoubleSine     (b) Counter Example     (c) Himmelblau     (d) 10D-Rastrigin

Figure 5: Plots of all the synthetic objectives used in the experiments. We have used a 10-dimensional Rastrigin function and the figure in (d) is a two-dimensional one.

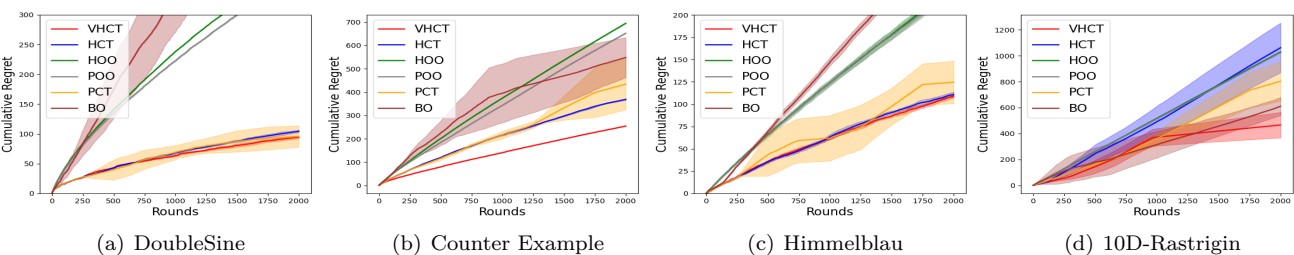

(a) DoubleSine     (b) Counter Example     (c) Himmelblau     (d) 10D-Rastrigin

Figure 6: Cumulative regret of different algorithms on the synthetic functions.

all the algorithms on the 10-dimensional space $[-1, 1]^{10}$. The results are shown in Figure 6(d).

$$f(\mathbf{x}) = -An + \sum_{i=1}^{n} \left[ A \cos\left(2\pi x_i\right) - x_i^2 \right] \text{ with } A = 10.$$

As can be observed in all the figures, VHCT is one of the fastest algorithms, which validates our claims in the theoretical analysis. We remark that Himmelblau is very smooth after the normalization by its maximum absolute value on $[-5, 5]^2$ (890) and thus a relatively easier task compared with functions such as Rastrigin. DoubleSine contain many local optimums that are very close to the global optimum. Therefore, the regret differences between VHCT and HCT are expected to be small in these two cases.

### E.3 Performance of VHCT in the High-noise Setting

Apart from the low-noise setting, we have also examined the performance of VHCT in the high-noise setting. In the following experiments, we have set the noise to be $\epsilon_t \sim \text{Uniform}(-0.5, 0.5)$. Note that the function values are in $[-1, 1]$, therefore such a noise is very high. As discussed in Section 4.4, it should be expected that the performance of VHCT is similar to or only marginally better than HCT in this case. As shown in Figure 7, the performance of VHCT and HCT are similar, which matches our expectation.

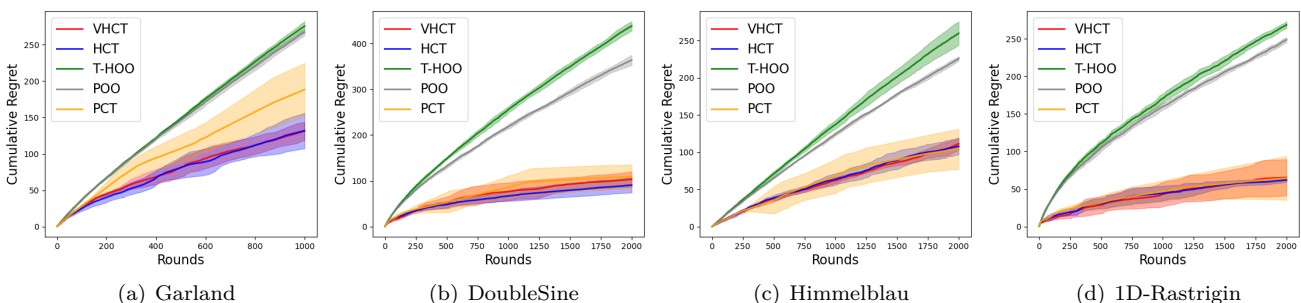

(a) Garland      (b) DoubleSine      (c) Himmelblau      (d) 1D-Rastrigin

Figure 7: Performance of different algorithms on the synthetic functions with high noise

