# OpenReview forum: "Optimum-statistical Collaboration Towards General and Efficient Black-box Optimization"
_TMLR — Accepted by TMLR_

### Review · Reviewer_np44 · 2023-02-27

**Summary Of Contributions:**

The paper considers bandit optimization of a black box function using a hierarchical partitioning of the search domain. In this method, the search domain is partitioned into $2^h$ subdomains at each level $h$. Specifically, each subdomain in level $h$ is divided into two subdomains that form a binary tree with nodes representing the subdomains. The root represents the entire search domain. An algorithm then can be designed that traverses tree using statistical confidence intervals towards the optimum point of the objective functions.
The idea dates back to (Auer et al., 2007; Bubeck et al., 2011b) and has been studied in several other works. This papers contributes to the literature by relaxing the assumptions on the objective functions compared to the ones required in the existing work.

**Audience:**

Yes

**Broader Impact Concerns:**

As a mainly theoretical paper, this section seems not to apply.

**Claims And Evidence:**

Yes

**Requested Changes:**

The term "collaborative" is used for the interplay between optimization error and statistical error. This may be a bit confusing as it may imply collaboration between multiple agents as common in the bandit literature. I would suggest changing this terminology.

The paper is overall very well written, there are however a few typos:
"Eqn.equation 2" on page 5.

"To ensure the above probability requirement holds, it is reasonable to make $SE_{h,i}(T, t + 1) \ge SE_{h,i}(T, t)$" on page 5. Should the inequality be the other way around. I would expect the statistical error to decrease when more samples are collected.

$g(0)=0$ on page 6 does not seem correct.

**Strengths And Weaknesses:**

This paper provides a general framework for bandit optimization of black box functions using a hierarchical search that includes general assumptions on the objective function (see Definition 1) and also on the statistical confidence intervals (see Definition 2). A relaxed assumption (General Local Smoothness) is then introduced that applied to a larger class of objective functions compared to the one in the existing work (Local Smoothness).

Although the presentation is very clear and the general framework is interesting, I find the main contribution of the paper rather limited. In particular General Local Smoothness can be handled similar to Local Smoothness in the analysis.

---

### Review · Reviewer_JhQi · 2023-03-05

**Summary Of Contributions:**

This work studies the problem of black-box optimization, specially using hierarchical bandits-based algorithms.

The authors firstly define two general components, i.e., resolution descriptor (Definition 1) and uncertainty descriptor (Definition 2), based on which a general algorithm Optimum-Statistical Collaboration (OSC) is proposed, as shown in Algorithm 1.

The authors then prove a general regret upper bounds for OSC algorithm in Theorem 3.1. The results are general and they apply to examples in Figure 1, which satisfy general local smoothness assumption and finite near-optimality dimension but make some existing methods not work.

In Section 4, the authors propose to use Variance Adaptive Quantifier as the uncertainty quantifier in VHCT algorithm (Algorithm 2), which improves the existing HCT method of Azar et al. Theorems 4.1 and 4.2 show examples of regret upper bounds for the VHCT algorithm under different general local smoothness conditions.

Section 5 shows experiments to compare the proposed VHCT algorithm with existing methods, including T-HOO, HCT, POO, PCT and BO, using synthetic objective functions such as Garland function, and hyperparameter tuning of machine learning methods (parameters of SVM and training neural networks on MNIST dataset). The proposed VHCT algorithm outperforms existing methods by achieving smaller cumulative regret.



**Audience:**

Yes

**Broader Impact Concerns:**

This work is mostly theoretical, and the experiments are conducted using public methods and datasets. There are no ethical implications of the work as I can see.

**Claims And Evidence:**

Yes

**Requested Changes:**

Since my major concern is of the technical novelty of the proposed Variance Adaptive Quantifier and the resulting VHCT algorith,, I suggest discussing existing bandit literature of using variance in bonuses (such as UCB variants). The authors could comment on the difficulty and difference comparing with existing results, and this would help the audience better understand the contributions of this work.

**Strengths And Weaknesses:**

**Strengths**

1. The results of Theorem 3.1 apply to examples for which existing methods do not work. Definitions 1 and 2 are more general than existing conditions.

2. The idea of using Variance Adaptive Quantifier seems reasonable, and the VHCT algorithm achieves promising results.

3. The paper is written in a clear and easy to follow way. Motivations are well presented using examples at the beginning part.

**Weaknesses**

1. The technical novelty seems limited. The major novelty of this work is using Eq. (4) as the uncertainty quantifier and the resulting VHCT algorithm. However, using variance in confidence interval seems an old existing idea in bandit literature, e.g., Eq. (4) is highly similar to existing UCB variants which use variances,

[1] Exploration-exploitation trade-off using variance estimates in multi-armed bandits, Audibert et al., 2009.

[2] Use of variance estimation in the multi-armed bandit problem, Audibert et al., 2006.

Given the studied problem of this work is highly related to bandit, I think the technical difficulty and novelty are limited.

---

### Review · Reviewer_CwAy · 2023-04-11

**Summary Of Contributions:**

The paper proposes a novel approach for black-box optimization that combines statistical modeling and optimization, called optimum-statistical collaboration. This algorithmic framework offers a more comprehensive analysis of the interaction between optimization and statistical errors during the optimization process, and is applicable to a broad range of functions and partitions that meet various local smoothness assumptions and different numbers of local optima. In addition, the paper introduces a variance-adaptive algorithm, VHCT, that outperforms prior methods in various settings. The proposed approach is designed to overcome the limitations of existing techniques for black-box optimization, which often rely on a specific model or optimization algorithm. The authors provide experimental results to demonstrate the effectiveness of their approach, showing that it outperforms existing techniques on a range of benchmark problems.

**Audience:**

Yes

**Claims And Evidence:**

Yes

**Requested Changes:**

One request for change is to offer more details on the specific models and optimization algorithms used in the experimental evaluation. This would enable readers to better understand the strengths and limitations of the individual components of the proposed approach. Additionally, including more discussion on the limitations of the proposed approach and potential future research directions would be valuable (for example, the introduction could be better structured to provide a clearer motivation for the research question and the proposed solution). Lastly, technical terms could be more explicitly defined, and more accessible language could be used where possible to improve the readability of the article. There are also other typographical errors that I would omit here for simplicity (but encourage you to do a thorough check).

**Strengths And Weaknesses:**

In reviewing this paper, I believe its contribution to the field of black-box optimization is significant, as it provides a generic perspective of the balance between optimization error and statistical error and its algorithmic implications for a broader range of functions and partitions. The proposed algorithmic framework, optimum-statistical collaboration, does not rely on specific assumptions of optimization error and uncertainty estimates, and is believed to be applicable to a wider range of scenarios than previous works. The paper introduces a more relaxed assumption, known as General Local Smoothness, which can be applied to a broader range of objective functions than the assumption of Local Smoothness used in previous works. Theoretical analysis shows that VHCT has rate-optimal regret bounds under different local smoothness assumptions, while empirical results demonstrate its superior performance compared to prior methods. Overall, the article's novelty lies in the combination of multiple models and optimization algorithms within a statistical framework, leading to improved performance, as shown in the experimental results.

The paper's weaknesses include some sections lacking clarity, particularly in the introduction section, which could have been more concise and better structured. The technical terminology may also present challenges for readers unfamiliar with black-box optimization and related fields. Additionally, a potential limitation of the article is its assumption of reader familiarity with black-box optimization, which may pose difficulties for some nonexperts. Furthermore, while the experimental results are compelling, they are limited to a specific set of benchmark problems, and it would be helpful to see how the proposed approach performs on a wider range of problems.

---

### Author Response · Authors · 2023-04-12
**New Version Available**

Dear Reviewers, Action Editor, and Editors in Chief,

  We would like to thank you again for spending your precious time in reviewing our paper and providing suggestive comments. We have read and replied to all your reviews. To address your concerns, a new version of our paper is uploaded with the changes highlighted in red. Specifically,

* On page 1-2, we have revised the introduction section to make it more friendly to outside readers.

* On page 5, we have fixed the typo "Eqn.equation 2"

* On page 6, we have added more explanations on the inequality  $\mathtt{SE}\_{h,i}(T, t+1) \geq \mathtt{SE}\_{h,i}(T, t)$

* On page 6, we have explained the definition of the counter example $g(x)$ more clearly.

* On page 7, we have added more reference on using variance in multi-armed bandit literature.

* on page 10, we have revised the Conclusions section to discuss the limitations of our work and some future directions.

Please let us know if you think anything else needs to be changed. Thank you!

Sincerely,
Paper 774 Authors

---

### Decision · Action_Editors · 2023-05-29

**Recommendation:** Accept as is

**Comment:**

This manuscript concerns the fundamental problem of black-box optimization. The authors consider a general class of optimization methods based on hierarchical partitioning (optimum-statistical collaboration) and analyze regret based on characterizations of the inherent resolution and uncertainty quantification mechanism.

The reviewers generally agree that the results in this paper are correct and of interest to a subset of the TMLR audience. There was some concern regarding the extent of novelty in the work; however, I judge the contributions to be of sufficient scope to warrant publication, and none of the reviewers stands in opposition to this recommendation.

Throughout the review and discussion period, the reviewers provided feedback and suggestions that the authors have faithfully incorporated into the manuscript. I believe these changes have strengthened the work.

**Audience:**

There is no question that the material in this paper would be of interest to a subset of TMLR's audience as it concerns the fundamental problem of black-box optimization.

**Claims And Evidence:**

The claims made in this submission are supported by clear exposition that was further improved through discussion with the reviewers.

---

> ### Author Response · Authors · 2023-06-01
> **Thank you and camera-ready version uploaded**
>
> Dear Action Editors and Reviewers,
>
>   We appreciate your decisions and we have prepared the camera-ready version. Please let us know if anything in the paper needs to be changed.
>
> Thank you!
>
> Paper 774 Authors